# A Colorful look at Climate Sensitivity

Bjorn Stevens[1] and Lukas Kluft[1]

[1]Max Planck Institute for Meteorology, Hamburg

**Correspondence:** Bjorn Stevens (bjorn.stevens@mpimet.mpg.de)

**Abstract.** The radiative response to warming, and to changing concentrations of $CO_2$, is studied in spectral space. If, at a particular wavenumber the emission temperature of the constituent controlling the emission to space does not change its emission temperature, as is the case when water vapor adopts a fixed relative humidity in the troposphere, or for $CO_2$ emissions in the stratosphere, spectral emissions become independent of surface temperature, giving rise to the idea of spectral masking. This way of thinking allows one to derive simple, physically informative, and surprisingly accurate, expressions for the clear-sky radiative forcing, radiative response to warming and hence climate sensitivity. Extending these concepts to include the effects of clouds, leads to the expectation that (i) clouds damp the clear-sky response to forcing, (ii) that diminutive clouds near the surface, which are often thought to be unimportant, may be effective at enhancing the clear-sky sensitivity over deep moist tropical boundary layers; (iii) even small changes in high-clouds over deep moist regions in the tropics make these regions radiatively more responsive to warming than previously believed; and (iv) spectral masking by clouds may contribute substantially to polar amplification The analysis demonstrates that the net effect of clouds on warming is ambiguous, if not moderating, justifying the assertion that the clear-sky (fixed RH) climate sensitivity – which after accounting for surface albedo feedbacks, we estimate to be about $3\,\mathrm{K}$ – provides a reasonable prior for Bayesian updates accounting for how clouds are distributed, how they they might change, and for deviations associated with changes in relative humidity with temperature. These effects are best assessed by quantifying the distribution of clouds and water vapor, and how they change, in temperature, rather than geographic, space.

## 1 Introduction

In recent years, conceptualizing the effects of thermal infrared radiation in spectral space has helped advance understanding of many basic aspects of Earth's energy balance and how it responds to forcing. For instance, a consideration of the differential spectral response of outgoing longwave radiation (OLR) to warming has proved crucial to understanding why OLR varies approximately linearly with temperature (Koll and Cronin, 2018), and how clear-sky radiative cooling is distributed through the depth of the troposphere (Jeevanjee and Fueglistaler, 2020; Hartmann et al., 2022). A spectral treatment of thermal-infrared radiation is also necessary to understand how radiation responds to forcing – in the form of increasing concentrations of atmospheric $CO_2$ (Wilson and Gea-Banacloche, 2012; Seeley, 2018; Jeevanjee et al., 2021b), and how it maintains an ability to respond to warming at very warm temperatures (Kluft et al., 2021; Seeley and Jeevanjee, 2021). All of the above studies

helped answer important questions by abandoning the idea that atmospheric radiative transfer could usefully be thought about as broadband, or grey.

The chief advantage of a grey atmosphere is heuristic. Conceptualizing the entirety of radiative transfer in terms of a single emission height, is a considerable simplification. In a grey world, intuition as to how the atmosphere responds to changes can be built around an understanding of what controls this emission height. This 'grey' way of thinking still greatly influences how we quantify changes to Earth's radiant energy budget, for instance when quantifying clear and cloudy-sky feedbacks. It turns out that thinking about radiative transfer more colorfully isn't that much more difficult, and by managing to do so it becomes possible to anticipate and quantify radiative responses to forcing[1] that 'grey thinking' either misrepresents or cannot explain. The chief simplification in treating the more colorful atmosphere is to recognize that different colors are controlled by different constituents, and to a good degree of approximation these constituents can be categorized as sensitive, or invariant emitters of thermal radiation. Quantification of their net effect, then follows quite simply from allowing invariant emitters to mask the response of sensitive emitters in proportion to their (the sensitive emitters) optical depth, something we call spectral masking.

The ideas presented here were developed in lectures on the greenhouse effect the first author gave at the Universität Hamburg, in the Fall of 2021. Many had their origins in joint work with the second author. Subsequently we became aware that others were, or had been, thinking along similar lines, to understand cloud-free atmospheres. For instance, the simple model of $CO_2$ forcing discovered and presented in those lectures had been found independently, and much earlier, by Wilson and Gea-Banacloche (2012), and has since been elaborated upon further and more thoroughly by Seeley (2018), Jeevanjee et al. (2021b), and Romps et al. (2022). Likewise, the ideas related to the clear-sky radiative response were being developed independently by Jeevanjee et al. (2021a); McKim et al. (2021); Colman and Soden (2021); Koll et al. (2023). In retrospect these studies do much of the heavy lifting that some readers would like to see by way of justifying some of the approximations we make. This allows us to focus on showing how this colorful way of thinking can be condensed into a heuristic that helps us think about climate sensitivity, and the role of clouds, more broadly. In this sense our work is less intended as a replacement for rigorous treatment of radiative transfer, and more as a way to understand the results of such computations.

The outline of the paper is as follows, after introducing the data sources and community tools used, the basic ideas are introduced in §3. These are used to derive estimates and provide understanding of Earth's clear-sky climate sensitivity and its components in §4. This provides a basis for thinking about Earth's equilibrium climate sensitivity more broadly (§5), and for better understanding the role of clouds in its determination. Conclusions and an outlook are presented in §6

## 2 Preliminaries

### 2.1 Data

Absorption spectra of selective absorbers, here $CO_2$ and $H_2O$ are taken from the catalog used for the Atmospheric Radiative Transfer Simulator, ARTS (Buehler et al., 2018; Eriksson et al., 2011). ARTS includes treatments of line broadening – with

---

[1]Here forcing is used generically, for instance to refer to a change of atmospheric composition, and distinguished from *radiative forcing*, which is the response.

the treatment of the foreign-broadening appropriate for Earth's atmosphere, and a representation of continuum absorption following the approach of Clough et al. (1989, 2005) as modified by Mlawer et al. (2012). Other data sources include monthly mean, gridded $(0.25° \times 0.25°)$ near surface $(2\,\text{m})$ air temperatures and column water vapor for the 240 months between 2001 and 2021, and are taken from reanalyses of meteorological data (ERA5, Hersbach et al., 2019). Cloud data is based on measurements using the AATSR instrument which flew on ENVISAT (Poulsen et al., 2019). The record extends from May 2002 through April 2012 and level 3 cloud-top temperature and cloud fraction are used.

## 2.2 Terminology and basic concepts

Concepts are developed for understanding the emission of terrestrial radiation, $99\,\%$ of which is emitted in the $50\,\text{cm}^{-1}$ to $2000\,\text{cm}^{-1}$ wavenumber (denoted by $\nu$) interval. This is sometimes referred to as the longwave or thermal infrared part of the electromagnetic spectrum.

We adopt terminology (see also Table 1) that will be standard for many readers. The Planck source function is denoted by $\mathcal{B}_\nu$, and depends on wavenumber, $\nu$ and temperature, $T$. The spectral irradiance is denoted by $F_\nu$ and unless indicated otherwise, is assumed to describe the outgoing thermal irradiance at the top-of-the-atmosphere. The mass absorption cross section $\kappa_{\nu,\text{x}}$ refers to a constituent 'x' (either 'c' for carbon-dioxide or 'v' for water vapor) whose density is denoted by $\rho_\text{x}$.

The optical depth between two heights, $z_1$ and $z_2$ is denoted by $\tau_{\nu,\text{x}}(z_1, z_2)$ and defined as

$$\tau_{\nu,\text{x}}(z_1, z_2) = \int_{z_1}^{z_2} \kappa_{\nu,\text{x}}\, \rho_\text{x}\, \text{d}z \approx \overline{\kappa}_{\nu,\text{x}} M_\text{x}(z_1, z_2) \tag{1}$$

The approximation defines the path integrated mass burden of x, denoted $M_\text{x}$, and a mean mass absorption cross section, $\overline{\kappa}_{\nu,\text{x}}$. Hereafter we denote the partial water vapor column burden, $M_\text{v}$ by $W$ and the partial $CO_2$ burden, $M_\text{c}$, by $C$. $W$ and $C$ are equal to their respective column burdens when the path is taken to extend through the entirety of the atmosphere. The effective mass absorption coefficient includes the effects of continuum absorption and pressure broadening by adopting a single value at an effective pressure and temperature, $(P, T) = (850\,\text{hPa}, 280\,\text{K})$. An exception is for the case of $CO_2$, as used in estimates of the forcing, for which we adopt values $(P, T) = (500\,\text{hPa}, 255\,\text{K})$, to be more representative of the levels where the forcing establishes itself. Reducing the effective pressure and temperature for $H_2O$ to $(P, T) = (700\,\text{hPa}, 270\,\text{K})$ changes estimates of the radiative response by about $2\,\%$.

The transmissivity through an absorber x is given as $e^{-\tau_{\nu,\text{x}}/\mu}$ where $\mu$ is the diffusivity factor. It is introduced by taking an effective zenith angle, $\theta$ to scale the path length by $\mu^{-1} = (\cos\theta)^{-1}$ through the medium, and thereby apply an equation originally valid for radiances, to irradiances. The value of $\theta$ depends on the optical depth (Armstrong, 1968), but a value of $\theta = 53°$ roughly corresponds to the average for optical depths uniformly distributed between 0 and 1, resulting in the commonly adopted value of $\mu^{-1} = 1.6$. and denoted $\mathcal{B}_\nu(T_\text{e})$, Beer's law thus becomes:

$$F_\nu(z) = \pi e^{-\tau_{\nu,\text{x}}(z, z_\text{e})/\mu}\, \mathcal{B}_\nu(T_\text{e}) \tag{2}$$

where subscript 'e' denotes the emission value of a particular variable, e.g., height, $z_\text{e}$ or temperature, $T_\text{e}$.

**Table 1.** Main symbols used in this study. Many are further specified by subscripts: e denoting emission, sfc denoting surface; cp denoting cold point; cs denoting clear sky; cld denoting cloud; v denoting vapor.

| Symbol | Meaning (units) |
|---|---|
| $\Lambda$ | Clear-sky longwave radiative response from heuristic model ($\mathrm{W\,m^{-2}\,K^{-1}}$). |
| $\eta$ | Efficacy of cloud masking of clear-sky longwave radiative response |
| $\kappa_\nu$ | Mass absorption coefficient at wavenumber |
| $\lambda$ | Sensitivity of broadband radiance to temperature $\partial_T F$ ($\mathrm{W\,m^{-2}\,K^{-1}}$). |
| $\lambda_{\mathrm{cld}}$ | Cloud contribution to $\lambda$. |
| $\lambda_{\mathrm{cs}}$ | Clear-sky contribution (broken into shortwave (sw) and longwave (lw) components) to $\lambda$. |
| $\mu$ | Cosine of effective zenith angle for radiance to irradiance conversion. |
| $\nu$ | Wavenumber, ($\mathrm{cm^{-1}}$). |
| $\rho$ | Density ($\mathrm{kg\,m^{-3}}$). |
| $\sigma$ | Stefan-Boltzmann constant ($\mathrm{W m^{-2} K^{-4}}$). |
| $\tau_\nu$ | Optical depth at wavenumber $\nu$. |
| $\chi$ | Fraction of spectrum (energy weighted) supporting the radiative response to warming. |
| $C$ | $CO_2$ mass burden ($\mathrm{kg\,m^{-2}}$). |
| $F_\nu$ | Spectral irradiance ($\mathrm{W\,m^{-2}\,cm}$), $F = \int F_\nu\,\mathrm{d}\nu$. |
| $N$ | Multiplicity of $C$. |
| $P$ | Pressure (Pa). |
| $T$ | Temperature (K). |
| $T_\star$ | Emission temperature in the absence of clouds and $CO_2$ (K). |
| $W$ | Water vapor mass burden ($\mathrm{kg\,m^{-2}}$). |
| $W_{\mathcal{R}}$ | $W(T)$ for fixed $\mathcal{R}$ at the given $T$. |
| $W_{\mathrm{sfc}}$ | $W(T_{\mathrm{sfc}})$, as fit to observations. |
| $f$ | Total optically thick cloud fraction. |
| $f_{\mathrm{h}}$ | 'High' cloud, defined as masking fraction of $CO_2$ forcing. |
| $f_\alpha$ | Effective cloud masking fraction of surface albedo changes ($\mathrm{cm\,kg\,m^{-2}}$). |
| $l$ | Slope of envelope of $15\,\mu\mathrm{m}$ $CO_2$ absorption band. |
| $z$ | Altitude (m). |
| $\mathcal{B}_\nu$ | Planck source function, depends on $\nu$ and $T$ ($\mathrm{W\,m^{-2}\,cm}$). |
| $\mathcal{F}(N)$ | Radiative forcing from an $N$-fold increase in $CO_2$, default value of $N = 2$. |
| $\mathcal{R}$ | Relative humidity. |
| $\mathcal{S}$ | Climate sensitivity (K). |

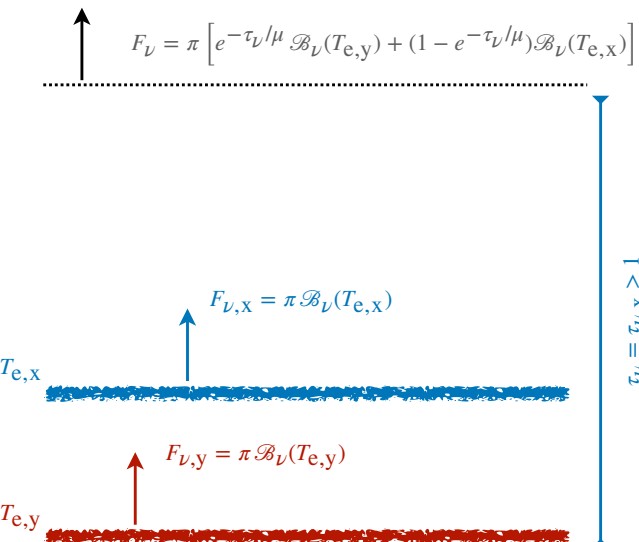

**Figure 1.** Schematic of simplified treatment of irradiances originating from two sources, denoted by $x$ and $y$ respectively, with each emitting as a black body ($\mathcal{B}_\nu$ denotes the Planck source function) from a height where their respective optical depths, $\tau_{x,y}$, (as measured from space) are unity. The factor $\mu$ in the tranmissivity (exponential terms) is the diffusivity factor that arises in converting radiances to irradiances.

## 3 Heuristics

Our colorful Ansatz amounts to the very simple, and rather standard, idea that emission to space at any given wavenumber is controlled by the emission temperature of the atmospheric constituent that first becomes optically thick at that wavenumber, and that emissions changes depend on how that absorber changes. We formalize this idea with the help of Fig. 1, which outlines how we smoothly weight the emissions from two absorbers (the lower one could be the surface) based on the optical thickness of the absorber which dominates the atmospheric emissions. Mathematically

$$F_\nu = \pi \left[ e^{-\tau_{\nu,\text{x}}/\mu} \mathcal{B}_\nu(T_{\text{e,y}}) + \left( 1 - e^{-\tau_{\nu,\text{x}}/\mu} \right) \mathcal{B}_\nu(T_{\text{e,x}}) \right] \tag{3}$$

where x is the dominant absorber and becomes optically thick at some temperature $T_{\text{e,x}} = T_1$. The second absorber, or possibly surface, emits at the temperature $T_{\text{e,y}} = T_2 > T_1$ at which it becomes optically thick. A simple variant of this model, one that perhaps better illustrates the way of thinking it formalizes, is the 'First to One' model[2], which simply replaces the transmissivity by zero or one depending on whether or not $\tau_{\nu,\text{x}} > 1$.

To help us understand how $F_\nu$ responds to changes in the surface temperature, $T_{\text{sfc}}$, thermal emissions at a given wavenumber are classified as arising from either a *sensitive* or $T_{\text{sfc}}$-*invariant* emitter.

---

[2]The name expresses the idea that the first absorber to have an optical depth of unity, as measured downward from the top-of-atmosphere, wrests control of emissions to space from the surface.

**Sensitive emitters** are ones whose emission temperature change with $T_{\mathrm{sfc}}$, such that $\delta T_{\mathrm{e,x}} = \gamma \delta T_{\mathrm{sfc}}$ with $\gamma > 0$ a proportionality constant.

**Invariant emitters** are ones whose emission temperature is independent of $T_{\mathrm{sfc}}$, so that $\delta \mathcal{B}_{\nu,\mathrm{x}} = 0$.

The surface, at all wavenumbers, is an obvious example of a sensitive emitter, with $\gamma = 1$. At wavenumbers where it becomes
optically thick in the troposphere, $CO_2$ also behaves like a sensitive emitter. In that case, following a moist adiabat, $\gamma > 1$. Its precise value depends on how high in the troposphere its emission originates. To the extent the water-vapor path is only a function of temperature – something Jeevanjee et al. (2021a) call Simpson's law – it behaves as an invariant emitter. Likewise, to the extent the stratosphere adjusts its temperature to maintain radiative equilibrium, $CO_2$ emissions from the stratosphere acts as an invariant emitter.

The simple model, Eq (3), and concepts here introduced, are not intended as a replacement for radiative transfer modelling. Its purpose is mainly to formalize the selection of a dominant emitter at a given wavenumber, and show how this knowledge (when combined with the essential properties of that emitter) proves surprisingly informative of how irradiances will change with warming, or forcing, for instance as calculated by more complex models.

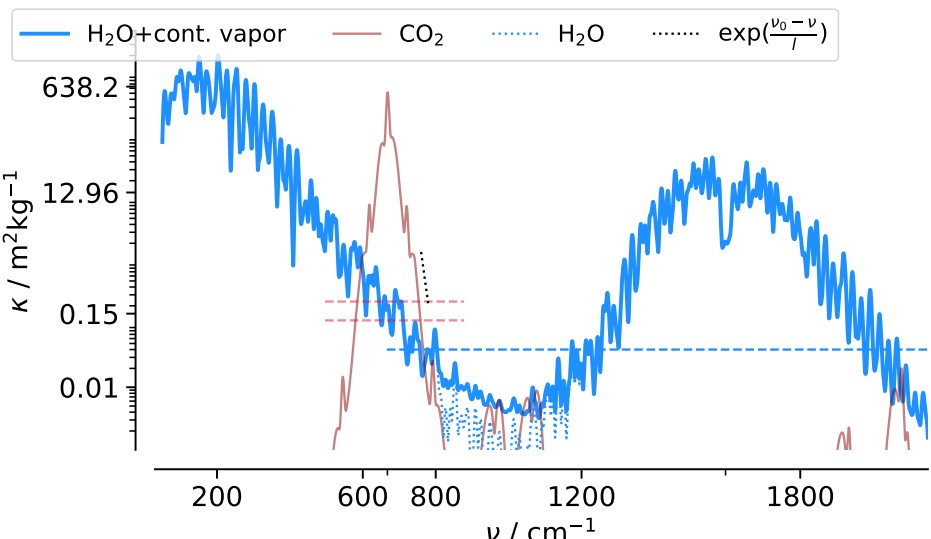

**Figure 2.** Mass absorption spectrum of $H_2O$ (blue) and $CO_2$ (red) as a function of wavenumber $\nu$. Spectra are calculated at a wavenumber interval of $0.05\,\mathrm{cm}^{-1}$ for a temperature of $280\,\mathrm{K}$ and pressure of $850\,\mathrm{hPa}$ and smoothed by convolving with a Gaussian ($9\,\mathrm{cm}^{-1}$) filter to show the absorption envelope. The black-dotted line ($l = 10.2\,\mathrm{cm}^{-1}$) is fit to the envelope of the $CO_2$ band, and the blue-dotted line shows the water vapor absorption in the absence of continuum absorption.

## 3.1 Spectral masking and the fractional support for the emission response

We introduce the idea of spectral masking, as a useful implication of combining Eq. (3) with our classification of emitters. To illustrate the idea we consider the case where water vapor is the only atmospheric absorber, so that $T_{e,y} = T_{sfc}$.

Accepting, for the moment, our assertion that the water vapor emission temperature remains invariant, it then follows from Eq. (3) that

$$\delta F_\nu = \pi e^{-(\tau_{\nu,v}/\mu)} \, \delta \mathcal{B}_{\nu,sfc}, \tag{4}$$

where $\delta \mathcal{B}_{\nu,sfc}$ denotes changes from surface emissions at wavenumber $\nu$. Eq. (4) can be derived more formally (see e.g., Eq. (5) in the SI of Koll and Cronin, 2018), which motivates Eq. (3) as a formalization of our ideas, instead of the simpler 'First-to-one' model. From Eq. (4), at wavenumbers where water vapor is optically thick $\delta F_\nu \to 0$. This is what is meant by spectral masking. Put more generally, at wavenumbers where an invariant emitter dominates emissions, it "masks" the radiative *response* of underlying, sensitive, emitters to warming. Jeevanjee et al. (2021a) call this spectral cancellation of surface feedbacks. We prefer the term masking, because the surface still responds to warming, but as viewed from space, the response is hidden, or masked.

The mass absorption cross sections of $H_2O$ and $CO_2$ are presented in Fig. 2. For $W \approx 25 \, \mathrm{kg \, m^{-2}}$, corresponding to the present day globally averaged column burden, at wavenumbers where $\kappa_{\nu,v} > 0.04$ the atmosphere is considered to be optically thick. This is satisfied over most of the thermal infrared, the exception being wavenumber between $800 \, \mathrm{cm^{-1}}$ to $1200 \, \mathrm{cm^{-1}}$, which defines the atmospheric window and emphasizes that it depends on the value of $W$. Fig. 2 also shows that $CO_2$, whose column burden $C \approx 6 \, \mathrm{kg \, m^{-2}}$, is the dominant absorber between $585 \, \mathrm{cm^{-1}}$ to $750 \, \mathrm{cm^{-1}}$, and will need to be accounted for in any fuller treatment of the radiative response to warming.

Because $W$ increases exponentially with $T_{sfc}$, the atmosphere will become opaque at lower values of $\kappa_{\nu,v}$ as $T_{sfc}$ rises, thus reducing its ability to transmit a radiative response to space. We quantify this effect through the introduction of a quantity

$$\chi(T) = \frac{1}{4\sigma T^3} \int\limits_0^\infty \frac{\mathrm{d}F_\nu}{\mathrm{d}T} \, \mathrm{d}\nu < 1, \tag{5}$$

which measures the broadband sensitivity of radiant energy to warming relative to that expected for a black body. Koll and Cronin (2018) introduce the same quantity (their Eq. (4)) and call it the average transmission. We prefer to think of $\chi$ as the fractional (spectral) support for the radiant response, in part because this terminology aligns better with the more colorful way of thinking, and the 'first-to-one model' that we keep in the back of our minds.

As an example, for the simple case of the water-vapor only atmosphere, $\delta F_\nu$ is given by Eq. (4) and $\tau_{\nu,v}(T) = \kappa_{\nu,v} W(T)$, such that

$$\chi = \frac{1}{4\sigma T^3} \int\limits_0^\infty e^{-\frac{\kappa_{\nu,v} W(T)}{\mu}} \left( \frac{\mathrm{d}\mathcal{B}_\nu}{\mathrm{d}T} \right) \mathrm{d}\nu.. \tag{6}$$

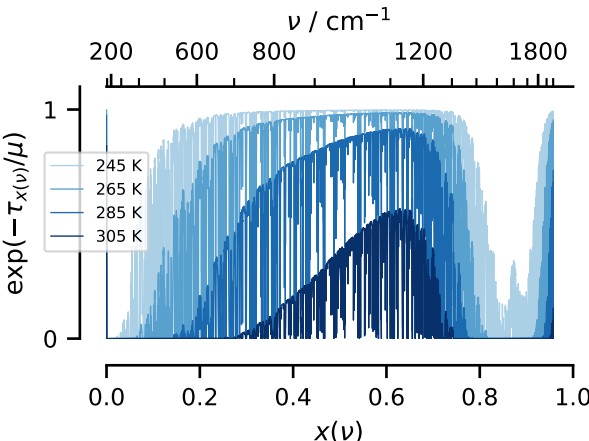

**Figure 3.** Spectral transmissivity plotted versus the cumulative black-body emission sensitivity, $x = (4\sigma T^3)^{-1} \int_0^\nu \left(\frac{d\mathcal{B}_{\nu'}}{dT}\right) d\nu'$. The corresponding wavenumbers are indicated along the upper scale. Line colors darken with $T_{\text{sfc}}$ with $W = W_{\mathcal{R}}(T_{\text{sfc}})$.

Rescaling $\nu$ by introducing the coordinate $x(\nu)$, such that

$$dx = \frac{1}{4\sigma T^3}\left(\frac{d\mathcal{B}_\nu}{dT}\right) d\nu \tag{7}$$

stretches the $\nu$-axis so that equally spaced $x$ intervals carry equal amounts of the radiative emission response to warming. In terms of $x$, $\chi(T) = \int e^{-\kappa_{x,\text{v}} W(T)/\mu} dx < 1$ is just the area under the curves in Fig. 3, and shows how an emission response is supported over some subset of $x$ corresponding to wavenumbers where water vapor is optically thin or transparent, i.e., $\kappa_{x,\text{v}} \ll \mu/W(T)$.

For the 'First-to-one' model, the curves in Fig. 3 would vary between zero and one. Intermediate values emerge both due to spectral averaging and from intermediate optical detphs. They highlight the complexity of the line-by-line variability of the spectral transmissivity, $e^{-\kappa_{x,\text{v}} W(T)/\mu}$ (which the stroke width used to render the plot is too wide to fully resolve). Effects of differences between near-line, versus continuum (or far-line/dimer), absorption, on $\chi$ can also be discerned by the way in which the window closes in Fig. 3. The former is associated with a narrowing of the window (region of support) with temperature, while the latter is apparent by weaker support as $W$ becomes large. Continuum emission is more broad-band or grey, whereas line-absorption, which more nearly results in $e^{-\kappa_{x,\text{v}} W/\mu} \in \{0, 1\}$, remains more colorful and better aligns with 'First-to-one' thinking (i.e., $\tau_{\nu,\text{v}}$ is either zero or much larger than one) and the concept of masking.

### 3.2 H₂O vapor – an invariant emitter

Simpson's law, provides the justification for idealizing water vapor in the troposphere as an invariant emitter, and hence Eq. (4). It states that if the relative humidity, $\mathcal{R}$, is fixed, $W$ depends only on $T$. Modulo effects of pressure broadening on $\kappa_\text{v}$, this means that $\tau_{\nu,\text{v}}$ likewise only depends on $T$, and hence the emission temperature (effectively where $T(\tau_{\nu,\text{v}} \approx 1)$) does not change with warming. This basic idea, was developed and used by a number of investigators to study runaway greenhouse at-

mospheres (Komabayasi, 1967; Ingersoll, 1969; Nakajima et al., 1992), before Ingram (2010) pointed out its earlier articulation by Simpson (1928).

### 3.2.1 Invariance of $W$ with $T$ with fixed $\mathcal{R}$

The statement that $\mathcal{R}$ does not change with warming (Arrhenius, 1896; Simpson, 1928; Manabe and Wetherald, 1967) contains a subtle ambiguity. Is $\mathcal{R}$, as a function of height, $z$, atmospheric pressure, $P$, or temperature $T$, constant as the surface warms? For a compressible atmosphere all three cannot be true, and which one is meant may have implications for Simpson's law. Assuming that $P(T)$ is bijective through the troposphere, whose top (or lowest pressure) is denoted by the cold-point temperature, $T_{\mathrm{cp}}$, it follows from the definition of $W$ that

$$
\quad W(T) \approx \int\limits_{T}^{T_{\mathrm{cp}}} P_{\mathrm{v}}(T') \left( \frac{R}{gR_{\mathrm{v}}} \frac{\mathrm{d}\ln P(T')}{\mathrm{d}T'} \right) \mathrm{d}T', \tag{8}
$$

with $R$ the mass specific gas constant for air, and $R_{\mathrm{v}}$ for water vapor alone. Here we neglect contributions to $W$ from the stratosphere, an assumption justified both by virtue of the smallness of $P_{\mathrm{v}}(T_{\mathrm{cp}})$ relative to its values at larger temperatures, and because we are mostly interested in $\mathrm{d}W/\mathrm{d}T$, which is constrained by the smallness of differences in the mass of the stratosphere as the surface warms. Simulations suggests that $T_{\mathrm{cp}}$ is effectively constant across a wide range of conditions characteristic of

175 the tropical atmosphere (Seeley et al., 2019). Hence we introduce it as a parameter, with the value $T_{\mathrm{cp}} = 194\,\mathrm{K}$ taken from radio occultation measurements in the tropics (Tegtmeier et al., 2020), bearing in mind that the same observations show substantially $(20\,\mathrm{K})$ larger values in the extra-tropics.

Eq. (8) establishes that $W$ depends only on $T$ as long as both $\mathrm{d}(\ln P)/\mathrm{d}T$, and $\mathcal{R}$, depend only on $T$. The former (a statement about the lapse-rate) is satisfied for an unsaturated adiabat, which well describes the temperature structure of the

180 upper troposphere. In the middle and lower troposphere, the temperature more closely follows the isentropic expansion of saturated air. The impact of allowing $\mathrm{d}(\ln P)/\mathrm{d}T$ to vary with $P$ as it would following a saturated adiabat, is illustrated by Fig. 4. It can be considerable in the lower troposphere. These profiles have been calculated for $\mathcal{R} = \mathrm{const.}$. Using a C-shaped profile of $\mathcal{R}$, as is more characteristic of the troposphere (Romps, 2014; Bourdin et al., 2021), albeit modified so the anchoring points depend on $T$, leads to similar conclusions. This then shows the extent to which Simpson's law, and many of

185 the idealizations that stem from its use, are limited by variation of $\mathcal{R}$ and $\mathrm{d}(\ln P)/\mathrm{d}T$ with $P$.

### 3.2.2 Observed variations of $W$ with $T_{\mathrm{sfc}}$

Over Earth's surface $W$ varies more weakly with $\mathcal{R}$ than it would were $\mathcal{R}$ held fixed, or if it were allowed to vary with $T$ as it does through the depth of the tropical troposphere. This is shown in Fig. 5 where we compare monthly averaged $W$ as a function of monthly averaged $T_{\mathrm{sfc}}$, which we denote $W_{\mathrm{sfc}}$. For a fixed $\mathcal{R}$, $W$ varies with $T$ following a different relation, which we denote by $W_{\mathcal{R}}$. Both vary exponentially with $T$, $W_{\mathcal{R}}$ more sensitively so. This enhanced sensitivity is robust to how

$\mathcal{R}$ is specified, so long as it remains constant with $T$; C-shaped profiles yield a similar slope. The relative flatness of $W_{\mathrm{sfc}}$ is

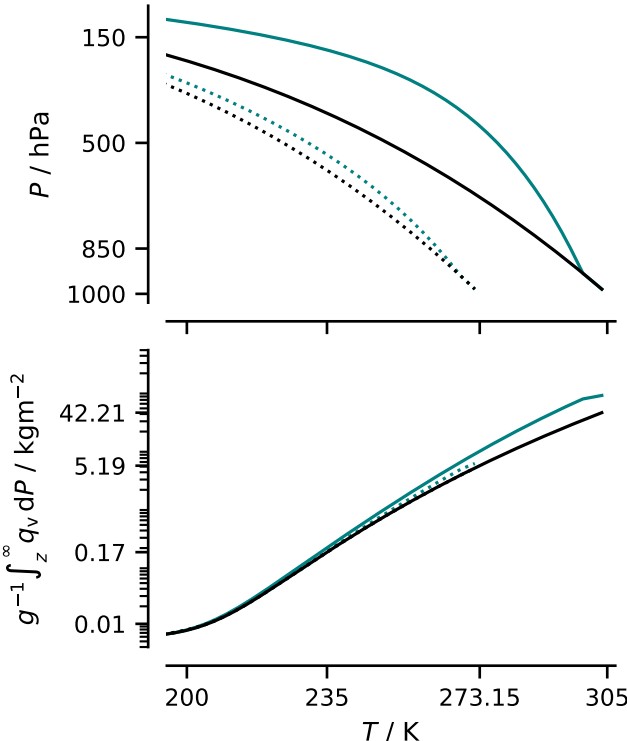

**Figure 4.** Theoretical temperature profiles and column humidities. Temperature profiles (top) following the formulation of the unsaturated (black) and saturated (teal) moist adiabats in Marquet and Stevens (2022) for two different surface temperatures (as indicated by the tick marks). Column water vapor, $W(T,$ between the top of the atmosphere and the height corresponding to the indicated temperature (bottom).

consistent with $\mathcal{R}$ being larger in the cold extra-tropics than over the warm sub-tropics, and is an imprint of the atmospheric circulation.

The implication is that the effect of the circulation is important for describing the spatial distribution of OLR and its scatter
(cf Fig. 1 in Koll and Cronin (2018)), for a given climate. But to the extent the circulation does not change strongly with warming, then $W_{\mathcal{R}}$ will better describe $W(T)$. In this case, with global warming one would expect the cloud of points in Fig. 5, to shift following $W_{\mathcal{R}}$ with global temperature changes. These findings motivate the rather simple choice of $\mathcal{R} = 0.8$, chosen so that $W_{\mathcal{R}}(T = \overline{T}_{\mathrm{sfc}})$ matches $W_{\mathrm{sfc}}(\overline{T}_{\mathrm{sfc}})$. A relative humidity of $0.8$ is larger than the mean $\mathcal{R}$, as it must be to capture the non-linearity of $W(T)$, whereby $\overline{W(T)} > W(\overline{T})$, with an over-bar denoting the global average.

**3.3    CO$_2$ gas – sensitive and an invariant emitter**

The heuristic formalized by Eq. (3) also helps understand how CO$_2$ influences the radiative response to warming. If, in radiative equilibrium, the absorption of radiant energy is independent of $T$, then the emission must also be independent of $T$. This is a rough description of the stratosphere, and means that at wavelengths where CO$_2$ is optically thick in the stratosphere, it behaves

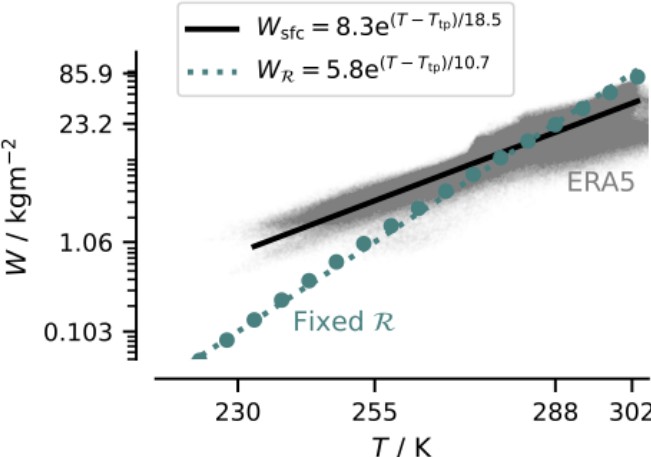

**Figure 5.** Monthly mean column water vapor, $W$, versus monthly mean temperature $T$; for $T = T_{\text{sfc}}$ (grey points); for the column defined between $T$ to $T_{\text{cp}}$, with fixed $\mathcal{R}(T)$ following an idealized C-shaped $\mathcal{R}(T)$ profile (filled teal-colored circles). Analytic expressions are fit relative to $T_{\text{tp}} = 273.16$ the triple point temperature, with a crossing point at present-day global temperatures. They are fit to the data by linearly regressing $\ln(W)$ binned by $T$.

like an invariant emitter.[3] This is not a consequence of Simpson's law, where concentrations adjust to temperature to maintain 205 the same emission. In this case, temperatures adjust to concentrations to maintain the same emission.

At wavenumbers on the shoulders of its central absorption feature (band), near $600 \, \text{cm}^{-1}$ and $733 \, \text{cm}^{-1}$, $CO_2$ is less absorbing, but still absorbing enough to become optically thick within the troposphere. At these wavenumbers $CO_2$ behaves like a sensitive emitter. In doing so it competes with $H_2O$ (more so at wavenumbers on the low energy side of the absorption band, where $H_2O$ is more absorbing, e.g., Fig. 2), for control of emission to space. At wavenumbers where $CO_2$ wins the battle, by 210 becoming optically thick above the emission height of water vapor, it re-establishes a radiative response to warming, that $H_2O$ would have otherwise masked. Where $CO_2$ emits at heights below the water vapor emission, its radiative response to warming is masked. The lack of concentration gradients in $CO_2$ complicate the picture, as they contribute to a more graduated change in $\tau_{\nu,\text{c}}$ than for $\tau_{\nu,\text{v}}$, which defocuses the emission height, and hence the idea of a single, or dominant, emitter.

Notwithstanding the difficulties of treating the overlap between $CO_2$ and water vapor at wavenumbers where both have 215 intermediate optical depths, Eq. (3) helps understand the basic physics of the radiant energy exchange, and anticipate effects that 'grey' thinking would obscure. Specifically, to account for $CO_2$ the dominant emitters in Eq. (3) are chosen based on whether or not an atmospheric absorber is optically thick at a particular value of $\nu$. When $\tau_\nu$ of one of the absorbers exceeded unity, its emission height and temperature are set to the height where $\tau_\nu = 1$. When both absorbers are optically thick, the dominant absorber is the 'first-to-one' (lowest emission temperature), and surface emissions (in that case, 'third-to-one') are

---

[3]Similar arguments could be applied to ozone, but its influence is not considered here.

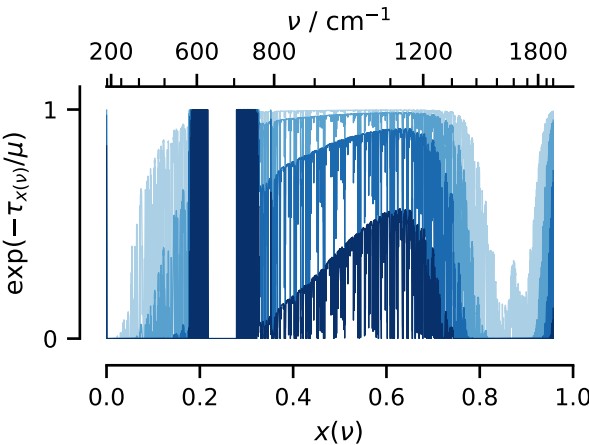

**Figure 6.** As Fig. 3, but accounting for the effects of $CO_2$ absorption.

neglected. By fixing the temperature of the stratosphere to $T_{\mathrm{cp}}$, we effectively account for stratospheric adjustment, and hence for the differentiated response of stratospheric versus tropospheric $CO_2$ to $\delta T_{\mathrm{sfc}}$..

Fig. 6 shows the fractional (spectral) support of the response, $\chi$, calculated using this model. In contrast to Fig. 3, which was calculated for water vapor alone, the spectral support for the radiative response vanishes in the vicinity of the central $CO_2$ absorption feature at $667\,\mathrm{cm}^{-1}$, and is re-established on its shoulders. Fig. 6 highlights the dual role of $CO_2$ in modulating the radiative response to warming. On the one hand, it masks surface emissions. On the other hand, it re-establishes a radiative response over parts of the spectrum that would otherwise be masked by water vapor. These effects depend on $T_{\mathrm{sfc}}$. The masking by stratospheric $CO_2$ becomes more important at colder temperatures, where the stratosphere is more massive, and the troposphere contains less water vapor. The re-establishment of the radiative response on the shoulder of the $CO_2$ absorption band becomes more prominent at larger $T_{\mathrm{sfc}}$, and is essential for maintaining some support for the radiative response at very warm temperatures. On balance, the presence of $CO_2$ moderates the dependence of $\chi$ on temperature (cf., Kluft et al., 2021; Seeley and Jeevanjee, 2021)

## 4 Spectral masking and the clear-sky climate sensitivity

In this section we apply our heuristic to help understand the radiative response to both warming and to forcing – the two ingredients of the clear-sky climate sensitivity. We show that Eq. (3) not only captures the conceptual content of this recent literature, but its prediction of the clear-sky sensitivity is also quantitatively accurate. This sets the basis for a understanding cloud effects in §5. There we show how clouds modify the clear-sky response in different ways, with a net effect that does not appear to differ substantially from zero. This establishes the expression for the clear-sky climate sensitivity as a useful estimate of the all-sky sensitivity.

## 4.1 Radiative response to warming

From our understanding of the temperature influence on the emission of thermal radiation, for small changes in $T_{\mathrm{sfc}}$, we expect

$$\delta F = \lambda \delta T_{\mathrm{sfc}}, \tag{9}$$

which introduces the proportionality constant, $\lambda$, as the radiative *response* parameter. It is closely related to the radiative *feedback* parameter, which is often denoted by the same symbol using the same expression, modulo a change in the sign convention to allow an increase in $F$ with $T$ to be associated with $\lambda < 0$, as expected for the net feedback in a stable system. In what follows we decompose $\lambda$ into a part that comes from changes in longwave and shortwave radiant energy transfer, such that $\lambda = \lambda^{(\mathrm{lw})} + \lambda^{(\mathrm{sw})}$.

In clear-skies, the longwave radiative response to a change in $T_{\mathrm{sfc}}$, as predicted by Eq. (3), with the 'first-to-one' approximation, is given by

$$\Lambda(T) \equiv \pi \int e^{-(\tau_{\nu,\mathrm{v}}/\mu)} \left( \frac{\mathrm{d}\mathcal{B}_\nu}{\mathrm{d}T} \right) \mathrm{d}\nu = \chi(T)\, 4\sigma T^3, \tag{10}$$

where we distinguish the radiative response estimated heuristically, which we denote by $\Lambda$, from the true value of clear-sky radiative response, which we denote by $\mathrm{cs}^{(\mathrm{lw})}$. For the case of a pure water vapor atmosphere, and modulo ambiguity in how $W$ is defined to vary with $T$, Eq. (10) is identical to Eq. (3) in Koll and Cronin (2018). It yields the expectation that

$$\lambda_{\mathrm{cs}}^{(\mathrm{lw})} \approx \Lambda(T_{\mathrm{sfc}}) = \chi(T_{\mathrm{sfc}})\, 4\sigma T_{\mathrm{sfc}}^3. \tag{11}$$

For $T_{\mathrm{sfc}} = 288\,\mathrm{K}$ and $\mathcal{R} = 0.8$, $\Lambda = 1.9\,\mathrm{W\,m^{-2}\,K^{-1}}$ (Fig. 7), which is indistinguishable from the McKim et al. (2021) estimate for $\lambda_{\mathrm{cs}}^{(\mathrm{lw})}$ under similar conditions. Kluft et al. (2019) estimate a slightly larger, $\lambda_{\mathrm{cs}}^{(\mathrm{lw})} \approx 2.3\,\mathrm{W\,m^{-2}\,K^{-1}}$, value, but this is consistent with their calculations having been based on a much drier atmosphere. Fig. 7 demonstrates that $\Lambda$ also captures the sensitivity of $\lambda_{\mathrm{cs}}^{(\mathrm{lw})}$ to temperature, humidity and the presence of $CO_2$, all forms of 'state-dependence' that have been identified and explored in a number of recent studies (Koll and Cronin, 2018; Bourdin et al., 2021; McKim et al., 2021; Kluft et al., 2021; Seeley et al., 2019).

The temperature sensitivity of $\Lambda$ is interesting in its own right, as it explains a state dependence of the climate sensitivity (see also McKim et al., 2021), here it is highlighted also because it will influence interpretations of cloud effects on the radiative response to warming. From Fig. 7 three temperature regimes can be identified. A cold, $T < 275\,\mathrm{K}$, 'Budyko' regime where $\Lambda$ is only slightly increasing ($\mathrm{d}\Lambda/\mathrm{d}T \approx 0.004\,\mathrm{W\,m^{-2}\,K^{-2}}$), and hence well approximated as constant. A warm regime, $285\,\mathrm{K} < T < 305\,\mathrm{K}$, over which the radiative response to warming reduces sharply, $\mathrm{d}\Lambda/\mathrm{d}T \approx -0.08\,\mathrm{W\,m^{-2}\,K^{-2}}$, with temperature. This is due to closing the atmospheric window by continuum absorption from water vapor (compared solid and dotted lines for $\chi$, likewise Fig. 3), and thus is sensitive to the humidity model $W_{\mathcal{R}}$ versus $W_{\mathrm{sfc}}$ (see also McKim et al., 2021, on this point). A third regime emerges at very warm temperatures, $T > 305\,\mathrm{K}$. Here $\Lambda$ is roughly constant, but small ($\Lambda \approx 0.25\,\mathrm{W\,m^{-2}\,K^{-1}}$). In this, regime $CO_2$ plays an important role in maintaining a radiative response (compare teal and black solid lines in Fig. 7) in an atmosphere that is optically thick in water vapor across the thermal infrared (Kluft et al., 2021; Seeley et al., 2019).

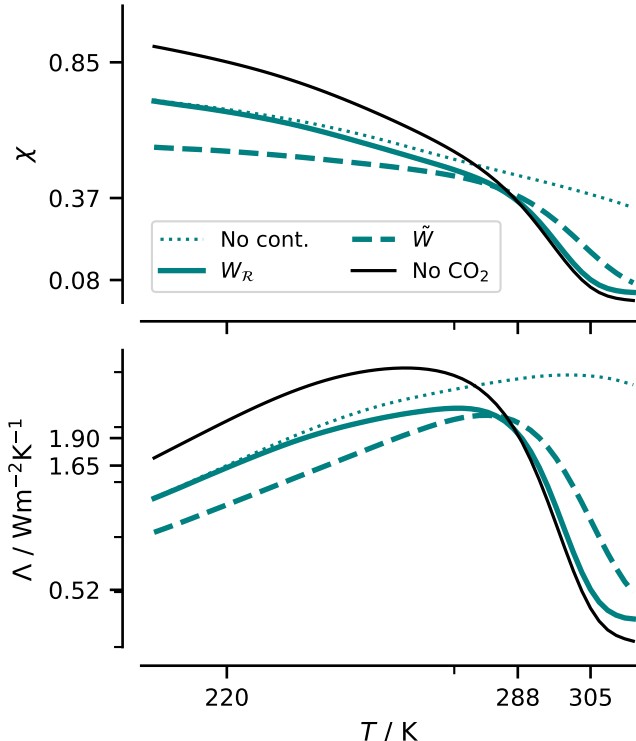

**Figure 7.** Variation of the support, $\chi(T)$, (upper) and the radiative response to warming, $\Lambda$ with $T$ (lower, minor $y$-axis ticks every 0.5) for different models of $W(T)$. Solid lines show calculations with the inclusion of continuum absorption the dotted line, for reference, shows the response in the absence of this absorption.

The moderating effects of $CO_2$ on the temperature dependence of $\Lambda$ reduces its maximum value from $2.55\,\mathrm{W\,m^{-2}\,K^{-1}}$ to $2.17\,\mathrm{W\,m^{-2}\,K^{-1}}$ and increases its minimum value from $0.05\,\mathrm{W\,m^{-2}\,K^{-1}}$ to $0.26\,\mathrm{W\,m^{-2}\,K^{-1}}$. The former effect arises from spectral masking at wavenumbers where $CO_2$ is optically thick within the stratosphere, and is more important in cold and dry atmospheres where surface emissions would otherwise dominate. The latter effect comes from $CO_2$ wing absorption

reclaiming spectral emissions from water vapor at warm temperatures (Fig. 6). The moderating effect of $CO_2$ on $\Lambda$ is somewhat smaller than the warm regime limit of $\lambda_{\mathrm{cs}}^{(\mathrm{lw})} \approx 1\,\mathrm{W\,m^{-2}\,K^{-1}}$ as estimated by Kluft et al. (2021) and McKim et al. (2021). Some of the difference can be explained by the use of an unrealistically cold stratosphere in those studies – decreasing $T_{\mathrm{cp}}$ to $150\,\mathrm{K}$ increases the asymptotic value of $\Lambda$ to $0.44\,\mathrm{W\,m^{-2}\,K^{-1}}$. The remaining difference likely reflects the crude treatment of emissions at intermediate optical depths by our model.

To the extent $\lambda_{\mathrm{cs}}^{(\mathrm{lw})}$ can be usefully approximated by $\Lambda(T_{\mathrm{sfc}})$, it demonstrates that this response is something that is quite easy to understand and, given knowledge of the $H_2O$ and $CO_2$ absorption spectra, to quantify. Moreover, because the dual effects of $CO_2$ appear to approximately compensate at Earth-like temperatures (see Fig. 7), $\Lambda \approx \Lambda_{\mathrm{v}}$. This indicates that the reduction in $\lambda_{\mathrm{cs}}^{(\mathrm{lw})}$ from what would be expected from a blackbody, largely measures how effective water vapor is at controlling emission to

space and thereby masking the spectral response of emissions to surface warming, an idea that Ingram (2010) seems to have been the first to appreciate. It also explains why simply approximating

$$\Lambda \approx \int\limits_{800}^{1200} \left(\frac{d\mathcal{B}_\nu}{dT}\right) d\nu,$$

(12)

as proposed by Colman and Soden (2021), and as might be justified by the 'First-to-one' model, provides such a reasonable estimate of $\lambda_{\mathrm{cs}}^{(\mathrm{lw})}$.

## 4.2 Clear-sky radiative response to (CO$_2$) forcing

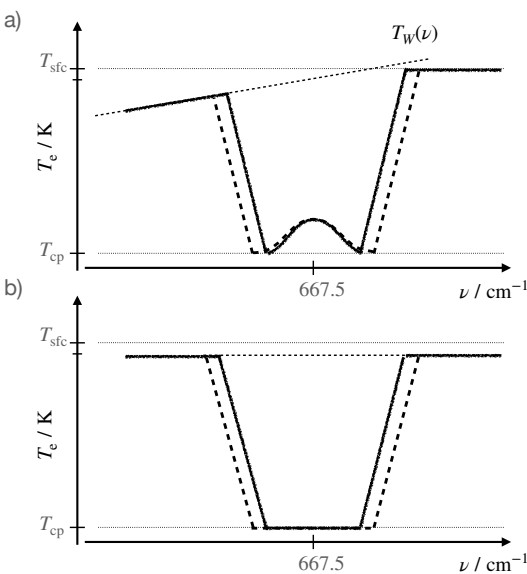

**Figure 8.** Schematic showing how CO$_2$ absorption is conceptualized a); and modelled (calculated), b). In a) stratospheric adjustment is conceptualized as maintaining stratospheric emissions near the line center at the same temperature. In b) An isothermal stratosphere (at $T = T_{\mathrm{cp}}$) models the invariance of CO$_2$ emission in the central part of the absorption band and the background water vapor emission is assumed constant across the band with its value at the line center.

Application of Eq. (3), yields a model of CO$_2$ forcing similar to that first proposed by Wilson and Gea-Banacloche (2012) and developed later, in more detail, by Jeevanjee et al. (2021b). The starting point is to describe the irradiance as a function of the CO$_2$ burden, $C$, its spectral mass absorption coefficient, $\kappa_{\nu,\mathrm{c}}$, and the limiting temperatures, $T_{\mathrm{cp}}$ and $T_\star$, such that

$$F(C) = \pi \int\limits_0^\infty \left[ e^{-C\kappa_{\nu,\mathrm{c}}/\mu} \mathcal{B}_\nu(T_\star) + \left(1 - e^{-C\kappa_{\nu,\mathrm{c}}/\mu}\right) \mathcal{B}_\nu(T_{\mathrm{cp}}) \right] d\nu,$$

(13)

With $T_\star = \min(T_{\mathrm{sfc}}, T_*)$ where $W_\mathcal{R}(T_*) = \kappa_{\nu,\mathrm{v}}^{-1}$. This defines $T_\star$ as the temperature at which the $W$, distributed with $T_{\mathrm{sfc}}$ following $W_\mathcal{R}$, would attain an optical thickness of one, or $T_{\mathrm{sfc}}$, which ever is smaller. Through its dependence on $\kappa_{\nu,\mathrm{v}}$ it will

vary with $\nu$. The choice of a fixed stratospheric $CO_2$ emission temperature set to the cold point (Fig. 8b) provides a simple way to account for stratospheric adjustment (Hansen et al., 1997), by ensuring that the emission temperature of stratospheric $CO_2$ remain invariant. As such, it anticipates our interest in the radiative response to changing $CO_2$, i.e., the forcing.

An $N$-fold increase in the burden, gives rise to a $\mathcal{F}$, given by the change in the irradiance (Eq. (13)) at the new burden, in clear skies this becomes:

$$\mathcal{F}_{cs}(N) = F(NC) - F(C) = \pi \int\limits_0^\infty \left( e^{-C\kappa_{\nu,c}/\mu} - e^{-NC\kappa_{\nu,c}/\mu} \right) [\mathcal{B}_\nu(T_\star)) - \mathcal{B}_\nu(T_{cp})] \, d\nu. \tag{14}$$

As climate sensitivity is usually referred to as the response to the temperature response of a doubling of atmospheric $CO_2$, in the remainder of the manuscript we equate $\mathcal{F}_{cs}$ with $\mathcal{F}_{cs}(2)$. With $T_{cp} = 200 \, \text{K}$ ranging from $194 \, \text{K}$ to $204 \, \text{K}$ (Tegtmeier et al., 2020), $\mathcal{F}_{cs}$ varies from $4.55 \, \text{W} \, \text{m}^{-2}$ to $4.22 \, \text{W} \, \text{m}^{-2}$. These values compare favorably with estimates of the adjusted clear-sky flux in the literature, which range from $4.3 \, \text{W} \, \text{m}^{-2}$ to $4.9 \, \text{W} \, \text{m}^{-2}$ (Kluft et al., 2019, 2021). The fidelity of this model is not only qualitative, but also quantitative as it captures the sensitivities to various quantities and as evident in more complex calculations, e.g., as in Jeevanjee et al. (2021b).

Following Wilson and Gea-Banacloche (2012) and subsequent studies, (e.g., Seeley, 2018; Jeevanjee et al., 2021b) two approximations make it possible to cast Eq. (14) into an even simpler form. The first is to replace the Planck source function with its band-averaged, or band-centered values. This is justified because the difference between the $CO_2$ transmissivities vanishes for $\tau_{\nu,c} << 1$ and for $\tau_{\nu,c} \gg 1$, so that $\mathcal{B}_\nu$ only contributes to the integral in the vicinity of $\nu_c$. This allows it to be approximated by its central value, and $T_\star$ to be approximated by a band averaged ($567.5 \, \nu$ to $767.5 \, \nu$) value,

$$\overline{T_\star} \equiv \frac{1}{200 \, \text{cm}^{-1}} \int\limits_{567.5}^{767.5} T_\star \, d\nu \approx 282.13 \, \text{K},$$

with $T_\star$ defined as previously described. The second approximation is justified graphically, from Fig. 2, which shows that the envelope of the $CO_2$ absorption spectrum falls off exponentially with $\nu$ as $\alpha e^{-\|\nu - \nu_c\|/l}$. This implies that for a $CO_2$ burden of $C$, $\tau_{\nu,c} > 1$ for $\nu_c - l \ln(\alpha C) < \nu < \nu_c + l \ln(\alpha C)$. It follows that for a burden of $NC$ the atmosphere becomes optically thick for the larger interval, larger by the amount $2l \ln(N)$. With these simplifications Eq. (14) simplifies to

$$\mathcal{F}_{cs}(N) \approx 2\pi l \ln N \left[ \mathcal{B}_{\nu_c}(\min(\overline{T_\star}, T_{sfc})) - \mathcal{B}_{\nu_c}(T_{cp}) \right]. \tag{15}$$

For the same range of $T_{cp}$ ($194 \, \text{K}$ to $204 \, \text{K}$), $\mathcal{F}_{cs}$ varies from $4.3 \, \text{W} \, \text{m}^{-2}$ to $4.0 \, \text{W} \, \text{m}^{-2}$, comparable to estimates from the direct integration of Eq. (14).

## 4.3 Clear-sky climate sensitivity

Dividing the estimate of the forcing from Eq. (14) by the radiative response from Eq. (10) gives an expression for the clear-sky climate sensitivity, $\mathcal{S}_{cs}$,

$$\mathcal{S}_{cs} = \frac{\int_0^\infty \left( e^{-\kappa_{\nu,c}C/\mu} - e^{-2\kappa_{\nu,c}C/\mu} \right) [\mathcal{B}_\nu(T_\star)) - \mathcal{B}_\nu(T_{cp})] \, d\nu}{\int_0^\infty e^{-\kappa_{\nu,v}W/\mu} \left( \frac{d\mathcal{B}_\nu}{dT} \right) d\nu} = 2.3 \, \text{K} \tag{16}$$

with $T_{cp}$ taken as the average across the stated range, and the net effect of $CO_2$ on the radiative response to surface warming assumed to be negligible. The additional simplifications of Eq. (15) for forcing, and Eq. (12) for the radiative response, yield a simpler expression in that it no longer depends explicitly on the absorption spectra of $CO_2$ and $H_2O$. With these approximations

$$\mathcal{S}_{cs} \approx \frac{\mathcal{B}_{\nu_c}(\min(\overline{T_\star}, T_{sfc})) - \mathcal{B}_{\nu_c}(T_{cp})}{2\sigma \int_{800}^{1200} \left(\frac{d\mathcal{B}_\nu}{dT}\right) d\nu} \, l \ln 2 = 2.4\,\mathrm{K}. \tag{17}$$

By virtue of assuming a fixed window, Eq. (17) will not, however, generalize as well as Eq. (16) to warmer temperatures.

As a comparison, for radiative convective equilibrium, Kluft et al. (2019) estimate $\mathcal{S}_{cs} = 2.1\,\mathrm{K}$ albeit for a drier atmosphere. The ability to derive Eq. (16) from the simple heuristic, and its interpretation/simplification in the form of Eq. (17) illustrates how the value of the clear-sky climate sensitivity, and its dependence on quantities like surface and tropopause temperature ($T_{cp}$), is quite easy to understand, and predict. This understanding, as we show next, provides a different, and we believe better, basis for quantifying the effect of clouds.

## 5  Inferences for Earth's atmosphere and estimates of the all-sky climate sensitivity, $\mathcal{S}$

In this section we explore how our more colorful way of thinking helps us understand how clouds influence the all-sky climate sensitivity, $\mathcal{S}$. Eq. (9), provides the basis for defining the climate sensitivity, $\mathcal{S}$ as the temperature response to a doubling of atmospheric $CO_2$, such that

$$\mathcal{S} = \frac{\mathcal{F}}{\lambda^{(lw)} + \lambda^{(sw)}}. \tag{18}$$

For a fixed planetary albedo[4] $\lambda^{(sw)} = 0$. In this case $\mathcal{S} = \mathcal{F}/\lambda^{(lw)} \neq \mathcal{S}_{cs}$. Which is to say that clouds influence the climate sensitivity through more than their effect on the planetary albedo

In §5.1 below, we explore how clouds influence $\lambda^{(lw)}$ and $\mathcal{F}$ independent of changes in cloud cover. We extend previous work that focused on cloud masking – what Yoshimori et al. (2020) called the cloud climatological effect – to show how changing cloud-top temperatures can actually enhance $\lambda^{(lw)}$ relative to $\lambda_{cs}^{(lw)}$. The impact of these effects are explored with a few examples in §5.2. In §5.3 we develop a framework for estimating $\mathcal{S}$, using estimates of cloud and surface albedo changes from the literature to calculate $\lambda^{(sw)}$, and link this to our understanding of $\lambda^{(lw)}$ to develop what we believe to a more physical framework for understanding how various processes influence $\mathcal{S}$, including the net effect of clouds.

### 5.1  The effects of clouds on the climate sensitivity for no changes in albedo

From a radiant energy transfer perspective, one important distinction between clouds and water vapor is that clouds are neither colorful, nor necessarily Simpsonian. Their greyness makes them effective in modifying both the clear-sky forcing, and the clear-sky radiative response to warming. Some of these effects are well known, but others are only beginning to be appreciated, or have been overlooked entirely.

---

[4]This implicitly also neglects changes in water vapor absorption with warming.

### 5.1.1 Cloud effects on forcing, $\mathcal{F}$

While it is well known that clouds mask the radiative forcing (Myhre et al., 1998), this is often overlooked when taking the measure of the cloud effect on climate sensitivity. For those wavenumbers where, in a cloud-free atmosphere, $CO_2$ controls the emissions to space, clouds with cloud-top pressures lower than $CO_2$ emission pressure, will wrest control of emissions, and mask changes from changing $CO_2$ concentrations. Even when cloud-top pressures are greater than the $CO_2$ emission pressure, so long as cloud-top temperatures lie below the clear-sky (and $CO_2$-free) emission temperature, $T_\star$, (see Eq.(14)) clouds will reduce the strength of the $CO_2$ forcing. Only in the case of clouds capping a surface inversion is it conceivable that they might increase $\mathcal{F}$ relative to its clear-sky value.

To quantify the reduction of cloud forcing from clouds, we define the high-cloud fraction to be the effective masking fraction, $f_{\mathrm{h}}$, such that

$$\mathcal{F} = (1 - f_{\mathrm{h}})\mathcal{F}_{\mathrm{cs}}. \tag{19}$$

It implies that, for $\mathcal{F}_{\mathrm{cs}} = 4.9\,\mathrm{W\,m^{-2}}$ (as calculated by Kluft et al., 2019), $f_{\mathrm{h}} \approx 0.25$ would result in $\mathcal{F} = 3.7\,\mathrm{W\,m^{-2}}$. To the extent that $f_{\mathrm{h}}$ should be compared to the geometrically high-cloud fraction, this appears to be a reasonable value. It is also consistent with Myhre et al. (1998) who estimate a similar, $27\,\%$, reduction in $CO_2$ forcing due to clouds.

### 5.1.2 Cloud effects on the longwave radiative response, $\lambda^{(\mathrm{lw})}$

When the cloud-top emission-temperature, $T_{\mathrm{cld}}$, does not change with warming, clouds mask window emissions in proportion to their (optically thick) cloud fraction (McKim et al., 2021), which we associate with the total (optically thick) cloud fraction, $f \approx 0.6$ (from AATSR). This leads to a nearly commensurate reduction in $\lambda^{(\mathrm{lw})}$, from its clear-sky value of $1.9\,\mathrm{W\,m^{-2}\,K}$ to $0.76\,\mathrm{W\,m^{-2}\,K^{-1}}$. We say 'nearly' because of the ability of $CO_2$ to restore some of the radiative response where its emission height lies above the clouds but below the tropopause. Because all clouds, rather than just high clouds, contribute to the masking of emissions from the surface, the reduction in the radiative response from cloud masking will be larger than the reduction of the forcing, roughly by a factor $(1 - f_{\mathrm{h}})/(1 - f) \approx 1.875$. This will increase $\mathcal{S}$ relative to $\mathcal{S}_{\mathrm{cs}}$, raising its value to $\approx 3.6\,\mathrm{K}$.

What seems to have escaped attention is how clouds might restore parts of the spectral response otherwise masked by water vapor. To quantify these competing effects, we model the effects of clouds on $\lambda^{(\mathrm{lw})}$ as

$$\lambda^{(\mathrm{lw})} \approx (1 - f)\Lambda(T_{\mathrm{sfc}}) + f\frac{\delta T_{\mathrm{cld}}}{\delta T_{\mathrm{sfc}}}\Lambda(T_{\mathrm{cld}}) = (1 - \eta f)\Lambda(T_{\mathrm{sfc}}) \tag{20}$$

with

$$\eta = 1 - \frac{\delta T_{\mathrm{cld}}}{\delta T_{\mathrm{sfc}}}\frac{\Lambda(T_{\mathrm{cld}})}{\Lambda(T_{\mathrm{sfc}})}. \tag{21}$$

If $\delta T_{\mathrm{cld}} = 0$ then $\eta = 1$ and Eq. (20) describes the masking of the clear-sky response (assuming $\lambda^{(\mathrm{lw})}_{\mathrm{cs}} \approx \Lambda(T_{\mathrm{sfc}})$, by clouds) – as discussed by McKim et al. (2021) and Yoshimori et al. (2020). The emission response across the spectrum as restored by clouds is manifest by $\eta < 1$; whereby $\eta < 0$, implying an all sky radiative response greater than that of the clear skies, is not precluded.

This demonstrates how the effect of clouds on the longwave radiative response depends on $\delta T_{\mathrm{cld}}/\delta T_{\mathrm{sfc}}$ through its effect on $\eta$. From Fig. 6 we can also infer that, for the same change in cloud-top temperatures, the ability to restore the radiative response will be stronger in the warm regime, where $\Lambda(T_{\mathrm{cld}})/\Lambda(T_{\mathrm{sfc}}) > 1$ than in the cold regime.

## 5.2 Some examples of cloud effects on the fixed albedo climate sensitivity

The above analysis identifies ways in which the amount and distribution of clouds influences estimates of climate sensitivity even if the coverage, albedo, and temperature of the clouds do not change. It also identifies $\delta T_{\mathrm{cld}}$ as a bit of a joker, through its ability to substantially increase or decrease the radiative response. Below we work through a few examples to illustrate these effects.

### 5.2.1 High clouds in the wet tropics

In the warm tropical atmosphere, where precipitating convection is embedded in a nearly saturated atmosphere (Bretherton and Peters, 2004), clouds may be especially important for the radiative response to warming. As the window closes, $\Lambda(T_{\mathrm{sfc}}) \to 0$, and there is little (only the $CO_2$ wing emissions) left for clouds to mask (Stephens et al., 2016). In this case the first term in Eq. (20) becomes negligible, independent of $f$, clouds with cold cloud-tops will carry the bulk of the radiative response, and its magnitude will be given by the second term, which is proportional to the cloud fraction and the cloud-top temperature change. This would provide a radiator for the tropical hothouse, one which together with wing emission from $CO_2$ (Kluft et al., 2021; Seeley and Jeevanjee, 2021) prevents the window from completely closing, thereby helping to moderate temperature increases. The degree of moderation will depend on the degree to which cloud-top temperature changes are constrained by the radiative cooling in the clear-sky atmosphere, which is still a matter of some debate (Zelinka and Hartmann, 2010, 2011; Bony et al., 2016; Seeley et al., 2019; Hartmann et al., 2022).

### 5.2.2 "Low clouds" coupled to surface temperature

In the case that clouds warm with the surface, $\delta T_{\mathrm{cld}} \approx \delta T_{\mathrm{sfc}}$, and $\lambda^{(\mathrm{lw})} \approx \Lambda(T_{\mathrm{sfc}}) + f\left[(\Lambda(T_{\mathrm{cld}}) - \Lambda(T_{\mathrm{sfc}})\right]$. In the warm regime $\Lambda$ decreases with temperature, and because cloud-tops are colder than the surface, $\Lambda(T_{\mathrm{cld}}) - \Lambda(T_{\mathrm{sfc}}) > 0$. Candidate cloud regimes for such behavior would be clouds topping the trade-wind layer (Schulz et al., 2021), or clouds in the doldrums. In these cases one might expect $T_{\mathrm{sfc}} - T_{\mathrm{cld}} \approx 7\,\mathrm{K}$ to $15\,\mathrm{K}$, with surface temperatures increasingly exceeding $300\,\mathrm{K}$. In this situation, from Fig. 7, clouds with tops at $288\,\mathrm{K}$ will radiate about a four-fold more energy per degree of warming than would a surface at $305\,\mathrm{K}$. More detailed calculations, e.g., Kluft et al. (2021), suggest a smaller, two-fold, difference, but suffer from simplifications to the stratosphere, suggesting that the real answer lies somewhere in between. In either case, the effect appears appreciable and illustrates how shallow boundary layer clouds, even small ones that cover most of the tropical oceans but generally go unnoticed (Mieslinger et al., 2022; Konsta et al., 2022), may help stabilize the climate. Over the cold extra-tropics, where $\Lambda$ increases with temperature, clouds (which emit at temperatures colder than the surface) have the opposite effect.

Measurements in the window region could help answer how much clouds warm with surface temperatures, here we ask how much they would have to warm to counter their additional masking effect relative to that of the forcing. This situation would be met with $\lambda^{(\mathrm{lw})} \approx f_{\mathrm{h}}\Lambda$. From Eq. (20), with $\eta = f_{\mathrm{h}}/f$, this is satisfied for

$$\delta T_{\mathrm{cld}} = \delta T_{\mathrm{sfc}} \left(1 - \frac{f_{\mathrm{h}}}{f}\right) \frac{\Lambda(T_{\mathrm{sfc}})}{\Lambda(T_{\mathrm{cld}})} \approx \frac{1}{2}\delta T_{\mathrm{sfc}}, \tag{22}$$

for $f_{\mathrm{h}} = 0.25$, $f = 0.6$ and $\Lambda(T_{\mathrm{sfc}})/\Lambda(T_{\mathrm{cld}})$ slightly less than one (from Fig. 7, corresponding to the warm regime).

### 5.2.3   Multi-layer clouds

This analysis can be generalized to clouds distributed over multiple layers, by working ones way down through the successive contribution of layers of non-overlapped clouds:

$$\lambda^{(\mathrm{lw})} = \Lambda(T_{\mathrm{sfc}}) \left[1 - \sum_i \eta_i f_i'\right] \tag{23}$$

where $f_i'$ denotes the cloud fraction for layer $i$ (increasing downward) that is not geographically masked by clouds at layers $j < i$, and $\eta_i$ indexes changes in cloud-top temperature.

### 5.2.4   Clouds and the clear-sky polar amplification paradox

From the point of view of the radiant transfer of energy in the thermal-infrared, the idea that the polar latitudes should warm disproportionately is a curious one, as the radiative forcing from a doubling of atmospheric $CO_2$ is proportional to $T_{\mathrm{sfc}} - T_{\mathrm{cp}}$, which is much smaller in the polar regions, and the radiative response to warming is, by virtue of the absence of water vapor to mask surface emissions, particularly large. Put differently, from our understanding of $\mathcal{S}_{\mathrm{cs}}$, for a fixed albedo and in the absence of lateral energy transport, the tropics should warm substantially more than the poles as $CO_2$ increases. This is less of a paradox when one considers the differences between the poles and the tropics, whether it be by virtue of surface albedo changes, or the decoupling of the polar surface from the polar atmosphere. Here we point out the potential for clouds to also cause a differentiated response of the cold poles, versus the warm tropics, to warming.

To do so we compare estimates of the local sensitivity, $\mathcal{F}/\lambda^{(\mathrm{lw})}$. We calculate $\lambda^{(\mathrm{lw})}$ following Eq. (20), using $W_{\mathrm{sfc}}(T_{\mathrm{sfc}})$, to calculate $\Lambda(T_{\mathrm{sfc}})$, and $W_{\mathcal{R}}$ to calculate $\Lambda(T_{\mathrm{cld}})$. This is an admittedly crude way to treat the variation of $W$ with height at different geographic regions, but using $W_{\mathrm{sfc}}$ for the cloud term as well does not change the answer appreciably. The albedo is kept constant and clouds are represented using three bounding cases: (i) $f = 0$, which renders clouds as transparent; (ii) $\delta T_{\mathrm{cld}} = \delta T_{\mathrm{sfc}}$, whereby clouds warm with the surface; and (iii) $\delta T_{\mathrm{cld}} = 0$, what one might call Simpsonian clouds. To calculate the forcing, $\mathcal{F}$, requires an estimate of the fraction of the forcing, $f_{\mathrm{h}}$, masked by clouds at different latitudes. We estimate this quite crudely, based on the fractional decrease of the cloud-top temperature (as taken from the AATSR data) relative to the temperature change through the troposphere as a whole:

$$f_{\mathrm{h}} = 1.9 \left(\frac{T_{\mathrm{sfc}} - T_{\mathrm{cld}}}{T_{\mathrm{sfc}} - T_{\mathrm{cp}}}\right) f. \tag{24}$$

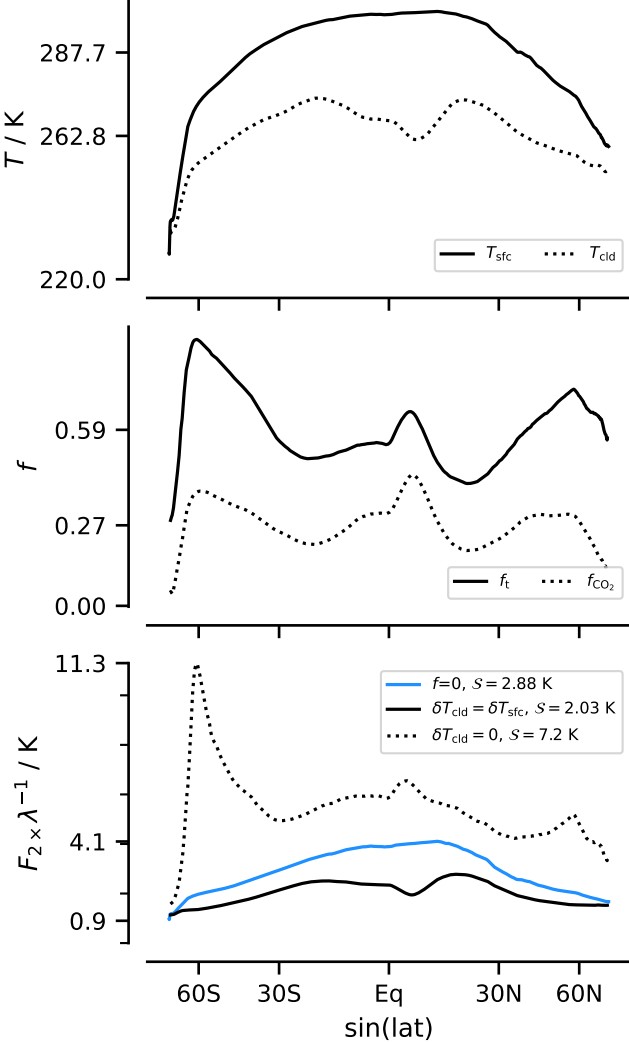

**Figure 9.** Latitudinal distribution of $T_{\mathrm{sfc}}$ and $T_{\mathrm{cld}}$ (upper), total cloud fraction $f$ and fraction assumed to mask $CO_2$ forcing, $\mathcal{F}$ (middle); and the ratio of the forcing $\mathcal{F}$ to the radiative response to warming, $\lambda^{(\mathrm{lw})}$ for different assumptions about clouds (lower).

The pre-factor (1.9) is introduced, and set, so that $\overline{\mathcal{F}}$ matches the estimate of $3.7\,\mathrm{W\,m^{-2}}$ of more detailed calculations. Because $\mathcal{S}$ is defined as a global (or statistical) quantity, it is estimated as $\overline{\mathcal{F}}/\overline{\lambda^{(\mathrm{lw})}}$.

The results of these calculations are shown in Fig. 9. For case (i), with transparent clouds, $f = 0$, values of $\mathcal{F}/\lambda^{(\mathrm{lw})}$ vary with latitude, from a low value ($0.9\,\mathrm{K}$) over the South Pole, to a high value ($4.1\,\mathrm{K}$) over the ITCZ region just north of the Equator, and thereby illustrating what we call the polar amplification paradox. For this case, $\mathcal{S} = 2.9\,\mathrm{K}$, which is slightly larger than the clear-sky estimates obtained previously using global mean quantities. For case (ii), with warming clouds ($\delta T_{\mathrm{cld}} = \delta T_{\mathrm{sfc}}$) $\mathcal{S} = 2.0\,\mathrm{K}$, with reductions most pronounced in the tropics, where additional emissions from clouds occurs in an atmosphere that is less masked by water vapor. Given the idea that high-clouds maintain a fixed temperature, this case might seem extreme, then again, warming along the moist adiabat is upward amplified, so that the case of fixed cloud height actually implies $\delta T_{\mathrm{cld}} > \delta T_{\mathrm{sfc}}$, which can be thought of as a form of lapse-rate feedback. For case (iii), with $\delta T_{\mathrm{cld}} = 0$, clouds mask the radiative response, and $\mathcal{S}$ increases considerably, inverting its geographic structure to be more poleward amplified. Hence, high-clouds that do not warm with the surface greatly sensitize the poles to increasing $CO_2$.

## 5.3 All sky climate sensitivity

Returning to Eq. (9), and introducing $\lambda_{\mathrm{cld}}$ to represent the (long and shortwave) radiative response to changes in the coverage (or albedo) of clouds and $(1 - f_\alpha)\lambda_{\mathrm{cs}}^{(\mathrm{sw})}$ to represent the all-sky changes in shortwave radiation with warming

$$\mathcal{S} = \frac{(1 - f_{\mathrm{h}})\mathcal{F}_{\mathrm{cs}}}{(1 - \eta f)\lambda_{\mathrm{cs}}^{(\mathrm{lw})} - \lambda_{\mathrm{cld}} - (1 - f_\alpha)\lambda_{\mathrm{cs}}^{(\mathrm{sw})}}. \tag{25}$$

By writing the surface albedo changes in terms of their clear-sky value, $\lambda_{\mathrm{cs}}^{(\mathrm{sw})}$, we explicitly account for cloud masking through $f_\alpha$, so that Eq. (25) explicitly accounts for the varied cloud effects on climate sensitivity (see also Table 2). The contribution of cloud coverage (or albedo) changes on the radiative response, $\lambda_{\mathrm{cld}}$ is usually associated with net albedo changes and historically has been the main focus of cloud feedback studies, the other terms are mixed together with the clear-sky response. To the extent clouds coverage/albedo changes are correlated with surface albedo changes $\lambda_{\mathrm{cld}}$, and $f_\alpha$ will not be independent. On a more detailed levels subtleties will arise due to differences in cloud albedo and cloud coverage; for instance, ambiguity among the terms may arise as clouds shift in location, and thereby changing the planetary albedo and their cloud-top temperature changes, while maintaining a fixed coverage.

Above it was shown that for $\delta T_{\mathrm{cld}} \approx {}^{1}/_{2}\delta T_{\mathrm{sfc}}$, we expect $\eta f \approx f_{\mathrm{h}}$. For clouds to maintain a 'neutral' effect on the climate sensitivity in the presence of cloud coverage changes would, from Eq. (21) with $\eta f \approx f_{\mathrm{h}} + \lambda_{\mathrm{cld}}/\Lambda(T_{\mathrm{sfc}})$, require

$$\delta T_{\mathrm{cld}} = \frac{f - f_{\mathrm{h}} + \lambda_{\mathrm{cld}}/\Lambda(T_{\mathrm{sfc}})}{f}\delta T_{\mathrm{sfc}}.$$

For $\lambda_{\mathrm{cld}} \approx -0.2\,\mathrm{W\,m^{-2}\,K^{-1}}$, as assessed by Forster et al. (2021), $\delta T_{\mathrm{cld}} \approx {}^{9}/_{14}\delta T_{\mathrm{sfc}}$. Fecent work suggesting that $\lambda_{\mathrm{cld}}$ may be even smaller (Myers et al., 2021; Vogel et al., 2022), motivates us to adopt this, admittedly crude, approximation. This amounts to approximating

$$\frac{(1 - f_{\mathrm{h}})\mathcal{F}_{\mathrm{cs}}}{(1 - \eta f)\lambda_{\mathrm{cs}}^{(\mathrm{lw})} - \lambda_{\mathrm{cld}}} \approx \frac{\mathcal{F}_{\mathrm{cs}}}{\lambda_{\mathrm{cs}}^{(\mathrm{lw})}} = \mathcal{S}_{\mathrm{cs}}. \tag{26}$$

**Table 2.** Principle cloud effects on climate sensitivity.

| Variable | Description |
| --- | --- |
| $f_h$ | Masking of $CO_2$ forcing |
| $f$ | Optically thick cloud fraction (masking clear-sky longwave radiative response) |
| $f_\alpha$ | Masking of clear-sky shortwave radiative response |
| $\eta$ | Efficacy of cloud masking of clear-sky longwave radiative response |
| $\lambda_{cld}$ | Net radiative response from changes in cloud coverage |

It then follows that

$$\mathcal{S} = \mathcal{S}_{cs}\left(1 - \frac{(1-f_\alpha)\lambda_{cs}^{(sw)}}{(1-f_h)\lambda_{cs}^{(lw)}}\right)^{-1} \approx \frac{4}{3}\mathcal{S}_{cs}. \tag{27}$$

The $4/3$ adjustment to the clear-sky climate sensitivity from surface albedo changes is estimated using the previously cited values of $f_h = 0.25$, with $f_\alpha = 0.5$, and $\lambda_{cs}^{(sw)} = 0.7$, from Pistone et al. (2014). Because the ice-margins are cloudier than the Earth as a whole, one might expect $f_\alpha > f$, however the complete masking of surface changes only arises for clouds with an optical thickness much greater than one. With $\mathcal{S}_{cs} = 2.3\,K$ this implies $\mathcal{S} \approx 3.07\,K$. Eqs. (16) and (27) point out how a reasonably physical, and quantitatively accurate, estimate of Earth's equilibrium climate sensitivity can be obtained by assuming that the main effect of clouds is to mask surface changes. And how, in this case the climate sensitivity can be reasonably estimated given knowledge of the $H_2O$ and $CO_2$ spectroscopy, which determines $\mathcal{S}_{cs}$, the total cloud fraction $f$ (as an approximation for $f_\alpha$), and an estimate of the surface albedo changes with warming.

For a planet without clouds, but with the same $\lambda_{cs}^{(sw)}$, $\mathcal{S} \approx 3.7\,K$, which is considerably larger. Turning the argument around, for a given $\delta T_{cld}$, this quantifies how large $\lambda_{cld}$ would need to be for clouds to make our planet more, rather than less, sensitive to forcing.

While an estimated climate sensitivity of about $3\,K$ will not raise any eyebrows, the way it was arrived at provides a new, and hopefully fertile, approach to thinking about clouds. Traditional feedback analysis adopts a grey perspective and attempts to explain sources of differences in estimates of $\lambda^{(lw)}$ due to changes in quantities such as the lapse-rate, or in humidity. This fails to adequately separate cloud from clear-sky effects, and obscures the essential question as to what controls the emission temperature of clouds, and how does their present-day distribution mask well understood clear-sky effects.

## 5.4 A new research programme for estimating $\mathcal{S}$

To better link the contributions of the radiative response to the physics of radiant energy transfer, a different research programme is needed. Such a programme would employ first-principle models of radiant energy transfer, and observations to:

1. quantify $\mathcal{S}_{cs}$ as the clear-sky Simpsonian response to warming, including the effects of $CO_2$ and other long-lived greenhouse gases (sensitive emitters);

2. quantify the contribution of cloud climatological effects, assuming clouds act as invariant emitters, i.e., the $f$, and $f_{\mathrm{h}}$ (assuming $\delta T_{\mathrm{cld}} = 0$) in the expression for $\eta$ in Eq. (25), to estimate what Yoshimori et al. (2020) call the cloud climatological effect;

3. quantify the corrections to $\lambda_{\mathrm{cs}}^{(\mathrm{lw})}$ from non-Simpsonian water vapor; to $\eta$ from non-Simpsonian clouds; and to $\lambda_{\mathrm{cld}}$ from changes to cloud coverage.

Koll et al. (2023) have taken steps to better quantify $CO_2$ effects on the $\mathcal{S}_{\mathrm{cs}}$ and the non-Simpsonian water vapor effects, but more is to be done. One strength of the proposed programme is that the first two steps can be constrained by theory and observations. Only the final step would require projections about future changes, or an extrapolation of past changes. If, in this step, the effects of clouds and relative humidity changes can be captured in terms of a few parameters, the method would lend itself well to Bayesian updating of those parameters, which could also be used to help quantify uncertainty.

## 6 Conclusions

We show that a simple heuristic that formalizes the control on emissions as a competition between two emitters, can explain both the radiative response to changes in long-lived greenhouse gases, and the response of clear-skies to warming. This makes it possible to derive an expression for the clear-sky climate sensitivity Eq. (16) and helps to understand and quantify state dependence, i.e., $\mathcal{S}_{\mathrm{cs}}$ increasing with temperature (Caballero and Huber, 2013; Bloch-Johnson et al., 2021) – increasingly so for $T_{\mathrm{sfc}} > 270\,\mathrm{K}$ – and with humidity at a fixed temperature (Bourdin et al., 2021; McKim et al., 2021).

Our heuristic provides a basis for thinking about how clouds modify $\mathcal{S}_{\mathrm{cs}}$. Even for no change in geographic coverage, clouds can both mask emissions from the surface, and restore what would have otherwise been a masked radiative response to warming. By virtue of locating at a different, usually colder, temperature than the surface, clouds that warm with the surface, amplify the radiative response over a warm surface (making the system less sensitive), and damp the response over a cold surface (making the system more sensitive). Clouds thus introduce an additional state dependence to the climate sensitivity, one that depends on the temperature of the underlying surface, and their own emission temperature. This state dependence renders estimates of $\mathcal{S}$ sensitive to not just how clouds change, but also their base-state distribution. It also means that Earth's geographic tendency to have more clouds where it is colder moderates geographic variations in the ratio of the local radiative forcing to the local response or thermal radiation, $\mathcal{F}/\lambda^{(\mathrm{lw})}$, and may thereby be a source of the poleward amplification of warming.

Some surprising properties of clouds that emerge from this way of thinking are: (i) the potential of diminutive clouds in the tropics, whose cloud top temperatures are more closely bound to surface temperature changes, to increase the radiative response of the tropical atmosphere to warming; (ii) the importance of even small cloud-top temperature changes in regions of deep convection for amplifying the radiative response of the moist tropics to warming; (iii) the importance of cloud masking at high-latitudes for increasing the sensitivity of regions whose clear-sky atmosphere would otherwise not be expected to be particularly susceptible to forcing. This highlights the many, albeit poorly quantified, ways by which clouds may reduce

the climate sensitivity. Small changes in cloud-top temperatures, or in the amount of very thin low clouds atop the tropical boundary layer can compensate or compound changes in optically thick clouds. This renders the *net* cloud contribution to warming ambiguous, and adds weight to the value of a theoretical understanding of the clear-sky climate sensitivity and the components which contribute to it.

When combined with estimates of surface albedo feedbacks from the literature, our heuristic can be used to quantify Earth's equilibrium climate sensitivity. The result, 3 K, doesn't meaningfully differ from values proposed by recent assessments adopting different approaches. However, our calculations are more transparently reasoned, and outline an observational programme to determine this number more precisely through: (i) estimates from the historical record how $\mathcal{R}$ is changing (cf Bourdin et al., 2021); (ii) estimates of cloud masking by quantifying their present distribution; and (iii) estimates of how cloud are expected to change with warming (in coverage and temperature) based on observed trends and symmetries. By parameterizing these effects the method would be amenable to Bayesian updating and uncertainty quantification.

This study emphasizes how corrections to the clear-sky climate sensitivity of a planet with fixed albedo is determined by the temperature of its clouds, how this temperature differs from the temperature of the surface, and how it changes. Observations, for instance by passive sensors sensitive to the most transparent parts of the spectrum or by active methods that can detect small and optically thin clouds (Wirth et al., 2009), that can help better quantify these corrections stand to advance understanding the most. Such measurements would help quantify the extent to which diminutive clouds, whose temperatures are coupled to the surface, strengthen the radiative response to warming, and by which high-clouds in cold regions, dampen it. Aligning the analysis of more complex models with the physics of the problem, e.g., by evaluating cloud responses in temperature and wavenumber, rather than in physical space, offers opportunities for gleaning more insight as to the plausibility of the processes these models simulate, or parameterize, and the ultimate role of clouds in modifying Earth's clear-sky climate sensitivity.

*Code availability.* The code used to produce all figures and make all calculations is provides as a Python notebook on Zenodo

*Author contributions.* The presented concepts and ideas have been developed by BS and LK during a joint lecture. BS has performed the analysis, created the figures, and written the original draft, and its revisions, based on comments raised by the editor and the reviewers and with input from LK.

*Competing interests.* We are not aware of any competing interests.

*Acknowledgements.* Jean-Louis Dufresne is thanked for encouraging the development of these ideas, and for his rigorous and thoughtful review which greatly improved the precision of the presentation. Manfred Brath helped with ARTS in the preparation and execution of the first author's greenhouse lectures. He and the developers of ARTS are thanked for their provision of such a useful community tool. Feedback

from the students and participants at special seminars at ETH-Zurich, the University of Bern, CFMIP in Seattle, and at the 2022 CERES team meeting in Hamburg, where these ideas were presented, is also acknowledged. Marty Singh helped clarify the first author's thinking on the Bayesian updating, and Nic Lewis helped identify, and clarify, some unclear points. Two anonymous reviewers and the editor, Paolo Ceppi, are also thanked for the time they spent in helping the authors clarify the presentation of their ideas. Nadir Jeevanjee and Daniel Koll, two pioneers of the lines of reasoning we also follow, are thanked for helping the authors better anchor their ideas in the rapidly developing

literature on this subject.

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
