# Peer review of "A Colorful look at Climate Sensitivity"

_EGUsphere, 2022_

## Author Comment (AC1)

**REPLY TO COMMENTS AND REVIEWS OF "A COLORFUL LOOK AT CLIMTE SENSITIVITY"**

BJORN STEVENS AND LUKAS KLUFT

**General Remarks – the good, the bad and the ugly**

On the "good" side, the paper was described as 'thoughtful, synthetic, provocative, containing,, stimulating new ideas and constituting a good read'. The "bad" could be classified as a lack of clarity in places and the neglect of compensating effects that muted some conclusions. The "ugly" was the concern that the manuscript 'repeated' too many recent developments, was in places 'overwrought', or unclear as to how results differed from the recent literature, or of the benefit of new terminology (spectral masking). Drawing encouragement from the good, we substantially restructured the manuscript to address the ugly, addressing the bad along the way. Some general points related to the "ugly" are noted first, as they apply to both reviews. An itemized response to the major points raised in each review is presented thereafter. We note that individual changes are presented in a cursory manner below, as difference between the original and revised manuscript pdf files are provided for this purpose.

*Repetition/Replication:* In rewriting the manuscript (especially §3 and §4) we make use of the existing literature to shorten the presentation where possible, and where results touch on previously published material, we mention these directly, foreshadowing this overlap in the introduction. These changes are most pronounced in §3. We considered to what extent this material could be deleted entirely, and at the risk of perhaps being a bit too attached to wanting to share this information in our own words, and with the excitement that we had upon uncovering it, we concluded that some degree of replication is inevitable, as one of the aims of the manuscript is to develop a unifying conceptual framework – spectral masking – for thinking about more varied contributions to Earth's climate sensitivity. As the reviewers point out this results in conclusions that are similar to has been published in recent work, in some cases ending up with analogous, or even identical equations. However without repetition, developing these ideas in ways that allow the reader to understand their extension to clouds, which is our most novel conclusion, would have been even more belabored, as we would have had to work backward to the reasoning. A few further, general points on this topic:

- The most onerous replication was the repetition of Eq. (4) from Koll and Cronin (2018), which we didn't realize, perhaps because their paper used this Eq. to emphasize the flatness of $\Lambda$ and we emphasize the variation, which is important for understanding the differentiated effects of clouds.
- Repetition with Jeevanjee et al., (2021) with regard to the clearsky feedback is now more directly acknowledged, although here too, while we come up with an equivalent result, as did Ingram much earlier, our way of getting there is different, and that is the point. This exemplifies our above assertion that a presentation which would start from Jeevanjee et al., would have to get the reader from their Eq. (13) to our (new) Eq. (5) in a way that conditions their thinking for what is to come. We thought it better to present the result

from the reasoning rather than the other way around, also because that makes it possible for readers without a deep understanding of the earlier literature to follow. That said, this motivation was not clear in the original manuscript, something our revisions address.

- Even if the repetition weren't necessary to make our broader point, or to establish the conceptual framework used to interpret clouds, it isn't disproportionate. Consider, for instance, the ratio of novel content to repetition in the recent papers on Simpson's Law, or the water vapor feedback as compared to what one learned from Nakajima et al., (1992) for the former, or Ingram's paper for the latter, likewise the degree of repetition in the recent work on $CO_2$ forcing relative to Wilson and Gea-Banacloche (2012).

**RC1**

The thoughtful, albeit critical, assessment of our manuscript, and especially the entreaty to more critically engage with the recent literature, spurred a thorough revision. While the reviewer might have imagined a different presentation of our ideas, we believe our revisions address the substance of the reviewer comment, and in so doing considerably improves the presentation. Below we address the major comments raised in the review, critical minor comments were all addressed, and noted as part of the response to the major comments. Of course we very much appreciated the positive minor comments, but don't inflate this response by mentioning them further.

We begin by responding to the introductory remarks raised in the review, in the form of their summary: *My overall feeling is that while this paper contains some useful syntheses and stimulating new ideas, it doesn't function very well in its current form.* We agree that our presentation style is a bit unconventional, and defend it as being reflective of our interest in showing how a particular way of thinking can help better frame the climate sensitivity problem, and give insight into clouds. The comments suggest that our initial attempt to make this point were not successful, something our revisions address. We did so by being more direct in linking the general idea (spectral masking) to the results, and better foreshadowing their implications for our more novel contributions, throughout the manuscript.. In this regard we also note that show that our way of reasoning could be used to replicate the results of others is actually desirable, something the more specific referencing now highlights.

**RC1 Major comments.**

(1) The authors acknowledge at the outset that they are replicating recent work, but argue that their ideas, as developed independently, are foundational for their more novel ideas about clouds. I did not find that to be the case, however. The material on the H2O feedback (sections 3.2-4.1) and CO2 forcing (section 4.2) seem to end up at the same place as previous authors, and I did not see anything new which was key to the later developments. I think readers would be better served by a more compressed review of recent findings, rather than a lengthy re-development. I expand on this in the next few items.

We revised the introduction to better explain our motivation for including this material. The presentation in §3.2-4.2 has been shortened, when results replicate earlier work this is pointed out more specifically, and justified either as a way to demonstrate how these results can be conceptualized using a common framework, or because they are necessary for what comes later. As an example, the section (§4.2) on $CO_2$ forcing – which recent papers devote an entire manuscript to – is now 3 equations and ca 20 lines. While it gets to the same place as Jeevanjee et al, it does so somewhat more simply (using the ideas of spectral masking),

albeit less rigorously. It become equivalent to Jeevanjee et al's Eq. (14) after applying the simplifications of Wilson and Gea-Banacloche (2012) which are summarized in only 12 lines (§4.3.1).

(2) Section 3.1 on the water vapor path $W$ seems overwrought. The theory in 3.1.1 seems very close to that of section 2 of the SI of Koll and Cronin, and could just be quoted as such. Also, while the comparison to observations in Fig. 2 is laudable, it feels unnecessary in a conceptual paper such as this. Furthermore, the difference in slopes between theory and obs in Fig. 2 is noticeable and goes entirely unexplained. This difference also results in the authors carrying around two forms of W(T) for the rest of the manuscript, despite the fact that the choice of W(T) doesn't seem to have much bearing on the results.

We keep this section because we needed the geographic distribution of $W$ for the treatment of cloud effects on the geographic distribution of radiative responses §5.1.4. The reason for the differences between $\tilde{W}$ and $W_{\mathcal{R}}$, which were only mentioned in passing in the original manuscript, are now better explained. Most other work also uses a fixed value of RH, or a dry adiabat (e.g., Kroll and Cronin, SI), which makes things simple (as we point out), but we wanted to establish for the reader how close this is to more reasonable distributions. Throughout we try, where possible, to connect our conceptual ideas to the empiricism, as in this case, or to more rigorous derivations, as for the case of $CO_2$ forcing. As a footnote, while our results (Fig 1, lower panel) for theoretical profiles is similar to Fig. 3 inset of Koll and Cronin (2018) their results essentially replicate Fig. 2 of Stevens and Bony, *Phys. Today* (2013), who in turn probably replicate results in earlier literature.

(3) The discussion of Simpsonian physics and its implications in 3.2 and 3.3 is nice, but the main results are virtually identical to those already found in the literature: Eq. 9 is equivalent to Eq. 13 of Jeevanjee et al. 2021a, and Eq. 10 here is actually identical to Eq. 10 of Koll and Cronin 2018. While these papers and others are cited in general in the introduction, they are not mentioned when these specific results are derived, so readers may wonder whether or not these results differ at all from those already in the literature.

We address this by connecting more directly to this literature, pointing out the similarity of our findings with what was found before. This however illustrates part of the challenge we faced. Had we relied on Koll and Cronin for our Fig 5, we would have been hard pressed to explain the warm and cold regimes, as their Figure was constructed in a way to emphasized the constancy of $\Lambda$, while for clouds its local maximum and lack of constancy is what is interesting. Because this might leave the reader wondering if the curves are similar, we add a quantitative comparison to show that this is indeed the case.

(4) I found Eq. 15 and its interpretation below Eq. 16 confusing. How would one derive this? As far as I can tell, it is an exact expression for the TOA flux for an isothermal stratosphere at temperature $T_{cp}$ overlying a surface with temperature $T_{sfc}$. I did not see a value-add to this section relative to existing treatments of simplified formulas for CO2 forcing (Wilson and Gea-Banacloche 2012, Jeevanjee et al. 2021b, Romps 2022 J. Clim.)

We discussed this above. In addition, in the revised manuscript we have adopted a different presentation: first we rearrange the initial expression to bring out the approximation being made to get the OLR; second we use the emission temperature at the outset (similar to Jeevanee et al., 2021). The former hopefully addresses the confusion, and better highlights the continuity of ideas developed in previous sections. The latter made it easier to address the overlap and allowed for a presentation that is both more precise and better rooted in the existing literature.

One advantage of using an isothermal stratosphere is that it effectively accounts for stratospheric adjustment (minor comment 5), also we now reference the Hansen work on this point.

(5) found section 5.1 to be the most provocative and stimulating of the paper. I appreciate this alternative approach to thinking about how clouds interact with climate feedbacks, as well as the underappreciated point that warming clouds will radiate through the window just as the surface does. But, the idea that the ratio of cloud radiation increase to surface radiation increase (denoted $\eta$) might differ significantly from 1 hinges on the exact shape of $\Lambda(T)$ in Fig. 5, which was derived under a variety of strong assumptions (no CO2, no pressure broadening, etc.). Furthermore, somewhat more comprehensive calculations, such as in McKim 2021 and Koll and Cronin 2018, show a reduced sensitivity of $\Lambda$ to temperature, with values plateauing near 2 $\mathrm{W/m^2/K}$ over a large range of $T_{\mathrm{sfc}}$. Indeed, the point of Koll and Cronin 2018 was to understand *why* $\Lambda$ varied so little with $T_{\mathrm{sfc}}$.

This point was addressed above. It is worth noting again however that the Koll and Cronin $\Lambda$ is not as flat as it looks. To convince ourselves we took a screenshot, blew it up, and measured it with the help of Adobe Illustrator. Their $\Lambda$ is only a little flatter, and this by virtue of RH=1 which lowers their peak. Our calculations include pressure broadening, in calculating the line-shapes, and always did. In the original manuscript we neglected the 'diffusivity' factor, this is now included (which meant revising most figures), and while it makes our $\Lambda$ better match that of Koll and Cronin, it does not change our conclusions or the degree of temperature dependence of $\Lambda$. The point that $CO_2$ lessens what would have otherwise been a heroic effort of dimunitive clouds is something we overlooked (Minor comment 6), and is now added as an important caveat where around line 334 we write: "This is a bit misleading however, . . . closer to 30 % . . ."

**RC2**

This is a thought provoking paper and a good read. The most novel and interesting part is Section 5. The earlier sections essentially repeat earlier work by Jeevanjee and Romps (2018) etc. and Koll and Cronin (2018) etc.. I think it would make a really useful paper if it is rewritten to bridge better to past literature and focus on what is really new.

This was a major focus of our revisions as explained above.

**RC2 Major comments.**

(1) Their spectral blocking approach is new phrasing. However, such approaches have a long history in the radiative transfer literature, e.g. correlated K codes and narrowband codes all split the spectrum into weak absorbing and strong absorbing spectral intervals - the tricky parts being those spectral intervals in between: which are often the same intervals that contribute most to forcing and feedback. I think the paper needs to say up front that we have radiative transfer models and radiative convective models that can solve things properly (at least the radiative transfer parts can be solved), and what the paper is developing a simple concepts to help understand models and observational results?

We agree, and have modified the introduction to address this point. Specifically we note that this old idea becomes particularly powerful when one is merely interested in explaining fluxes at the top of the atmosphere, and that the point of the manuscript is not to replace detailed radiative transfer, rather to say that conceptually it is easy to understand the results it produces.

(2) I found the radiative forcing section quite weak. It seems to be a less complete version of work by Jeevanjee, Huang and others - it also missed important concepts such as stratospheric and other adjustments which significantly alter the forcing. I think it would be better just to cite other work when building your climate sensitivivty arguments.

This has been reworked to better emphasize our goal of relating the derivation to the idea of masking. As noted in the response to RC1 one advantage of the isothermal stratosphere is that it effectively includes stratospheric adjustment, a point we now state explicitly.

(3) I'm not sure how much spectral blocking is really needed for the arguments. A paper that should be cited and contrasted with is Hartmann et al. (2022) https://doi.org/10.1175/JCLI-D-21-0861.1. They develop very similar argument using a more general spectral sensitivity concept.

As some of the cited literature shows, many of our results can be obtained in other ways. Our point is that spectral masking is a heuristic that can be used to help understand and anticipate the results of much more complex calculations. Which we now state directly two times (lines 31 and 86) for emphasis. Regarding the reference, we were not aware of this work, which we now cite.

(4) I wanted to learn from the cloud feedback section how concepts such a the increase in height on anvil clouds fair within the ideas presented - the same goes for decreases in stratocumulus decks. I felt that it would be useful to spell these out more explicitly.

We see this as part of the research programme that the manuscript articulates. The main point however, is that the focus should be less on the height and more on the temperature. To bring this out we have restructured the presentation of different cloud regimes in §5.

(5) I felt Section 5.3 on climate sensitivity was quite speculative and hand wavy. It might be better to explore what would be needed to make the cloud feedback large and positive - or negative? Also I was not persuaded by the overlap with albedo feedback. The albedo

feedback is mostly continental , whereas the SW cloud feedback is mostly over the oceans....

The numbers provided for cloud masking of the albedo feedback are taken directly from the cited literature. Having spent some time flying over the sea-ice margin along the Framm Strait, the first author can attest to the strength of cloud masking that this literature quantifies. We have modified the presentation on climate sensitivity to avoid the impression that our goal is to provide a new assessment, and rather to support the claim that the net cloud effect is more ambiguous than generallly appreciated, which adds weight to the value of the clear-sky estimates. In ongoing work being led by the second author, the outlined research programme is being adopted, as we work to quantify the effects of clouds more precisely using full radiative transfer.

---

## Referee Report (RR1)

Review of "A Colorful look at Climate Sensitivity" by Bjorn Stevens and Lukas Kluft

The authors have considerably revised and improved their manuscript, and I would like to thank them for that. This manuscript now has all the qualities needed to make an exceptional article, which will have a major impact on our community because of the advances it makes on the physical phenomena that control the value of climate sensitivity. I think that developing an approach to understanding, describing qualitatively but also quantifying climate sensitivity (i.e. forcing + response) was a goal for many of us, and it has now been achieved, at least for the LW. Although it is of course based on numerous previous studies, but bringing them together and making them consistent in order to quantify climate sensitivity, both under clear skies and with clouds, is an indisputable contribution. It seems to me that one of the challenges is to make this article readable by a wide audience, and therefore to make as little reference as possible to jargon, to make some additional comment when presenting results that are 'well known', but only to a small community. In the detail of the writing, it may be helpful to regularly remind people of the meaning of the different symbols used (forcing F, sensitivity S, etc..)

Major comments:
=============

1) While the work on all aspects of radiation in the LW is very detailed, the statements concerning changes in the cloud fraction on the one hand, and what is happening in the SW domain on the other, are treated superficially. The small change in cloud fraction (Myers et al., 2021; Vogel et al., 2022) only concerns tropical clouds. While the change in the liquid water content of clouds has little influence in the LW (and is therefore not discussed in this manuscript), it can have a much greater influence in the SW. For example, in mid-high latitudes, the SW effect of clouds is very different depending on how the water-ice transition is treated. In addition, section 5.4 is a bit of a "kitchen sink" in its current state. I think that part of this section should be elsewhere, in one (or more) section where the contribution of clouds to the 3 quantities, forcing, response, sensitivity would be formalised (see also my comment below on section 5.1). I understand the authors' desire to highlight the paradoxical nature of the role of clouds, but in this case their should consider a wider range of possible values and to be more explicit about what they have firmly established, what they 'roughly' estimate.

2) I feel some text is missing to explain how to read section 5.1 (clouds). Currently this section discusses the role of clouds on (1) forcing, (2) response (3) sensitivity, but this clarification is not done and only the "response" part is well structured. The effect of clouds on forcing is first presented very crudely (336-342), and then better formalised in the "polar" section, although this formalisation remains incomplete. For example, the term fCO2 is much discussed, but it is not precisely defined, and we do not know exactly where and how it comes into play. Why not having a section on "forcing" where all this would be clearly presented? The same for the sensitivity part, quickly and with little justification mentioned lines 343-344, then developed section 5.1.4 but without saying it explicitly.

In addition I'm still confused by the section on polar amplification. Based on results already presented in the manuscript, this section starts by explaining that the purely radiative sensitivity should be very high at the poles. Finally, using the equations themselves, in clear sky conditions, the authors find that the purely radiative sensitivity at the poles is low after all (Figure 9). So, what?

More specific comments (in text order):
=========================

Figure 1 is very welcome and very useful. However Figure 1, Eq. (3) and the text are not consistent. T1 and T2 are inverted between figure and text. I'm not sure T1 and T2 are necessary, Te,x and Te,y are probably sufficient. Equation of F in the figure and in the text are not consistent (I take Eq. 3 as the correct one). I'm not convince that the large arrows on the left (tau_nu,x >1) and on the right (tau_nu,y =1) are useful.

l. 158. Until now W was the amount of water vapour between the surface and the top of the atmosphere (line 75). From here you consider W as the amount of water vapour between TOA and where temperature is T. I think the distinction between the two is crucial to understand the manuscript.

l. 201: "... then the emission must also be independent of T. … explains why it cools when $CO_2$ levels rise." To write that the effect of $CO_2$ in the stratosphere is independent of temperature and in the following sentence to say that this explains the radiative cooling when $CO_2$ is increased will lose more than one reader, even if it is perfectly accurate. I suggest either deleting the sentence on cooling (which adds nothing here), or going into more detail.

l. 200-204: It seems to me that a major simplification of the reasoning is to neglect the radiative effect of ozone, when it is not, particularly in the LW. If I'm not mistaken, the entire stratosphere is in quasi-radiative equilibrium, so the adjustment doesn't just concern the spectral range where $CO_2$ absorbs, but also the ozone band at 9um. I have no problem with this simplification, but it deserves to be mentioned.

l. 221: "Fig. 6 shows χ…" => Fig. 6 shows the spectral transmissivity

l. 243: "… as predicted by Eq. (3)" and using the "First-to-one" approximation

l. 255: "We focus on the temperature sensitivity of Λ because it plays a role in interpreting the cloud effects…" but also to interpret climate sensitivity even without clouds!

Figure 7: please add the value corresponding to the lowest tick, it will help when reading line 262

l. 290: "where $W_R(T^*) = \kappa_{\nu,v}$ ," => where $W_R(T^*) * \kappa_{\nu,v} = 1$

l/. 290 .. as tbe => as the

l. 305 which we deonte => note(?)

l. 306: $T_W = 282.13$ K; How this value has been obtained?

l. 307: $\alpha e^{||\nu - \nu c||/l}$ : a minus is missing in the exponential

l. 309: "the amount $2l \ln(2)$"; the two "2" numbers have very different meaning and origin! I will suggest to consider a burden NC, therefore the amount will be $2l \ln(N)$, to avoid confusion and error. Same modification in Eq. 15

l. 327: Inferences for Earth's atmosphere and estimates of S => of sensitivity S

l. 337: "The degree of masking will mostly depend on the cloud-top pressure, although a more minor effect might arise if clouds set a colder baseline than water vapor, …" This sentence is hard to understand. I suggest to be more explicit, something like: If the cloud top pressure is lower than $CO_2$ emission pressure, then masking, otherwise not masking but setting TW in Eq. 14

l. 344: "from 2.2 W m $^{-2}$ K" why not 1.9 (results obtained with Eq. 11)?

l. 346: "...contribute to the masking of emissions from the surface" => from where radiation has been emitted, from where W is defined

l 347: please emphasise that $S\alpha$ is all sky sensitivity. Because $S\alpha$ is quite confusing, $S_{as}$ (all sky) could be more explicit.

l. 360: "magnitude";  add: (second term of Eq 18)

l. 360 thermostat => rather a radiator ?

328-330. beginning section 5 (and also in some discussions), I think it's important to specify that the cloud fraction is assumed to be unchanged

l. 397: … we compare estimates of the $S\alpha$ ;  …. the all sky sensitivity, $S\alpha$.  But why to compare $S\alpha$ with lambda, which is a sensitivity?

l. 402-404: I understand (I hope correctly) from Figure 9 (note from the text) that fCO2 is the "corrected cloud fraction" that masks the CO2 forcing. This should be made clearer. It should also be made clear how this affects the expression of forcing. There is probably a link with $\omega_F$ used later on. Why this parameter is introduce only here and not when discussing how to clouds affect the CO2 forcing (lines 336-341)? How the could temperature is computed is not specified (I think), and should be specified.

l. 419: Section 5.1.5: don't forget to specify that only the LW response of clouds have been considered here.

l. 420: … and temperatures do not change. What do you mean, as you analyse the response to a warming?

l. 442: … enough such that with $\omega_F < \omega_\lambda$ . Please put some words to explain/remember what $\omega_F$ and $\omega_\lambda$ are.

Some details:
==========

l. 73: "path integrated mass burden of x..." => "path integrated mass burden of x, Bx, ...". To avoid confusion with Planck, Mx could be better than Bx

l. 103: (?); ??

l. 146: ...the radiative response.=> the surface radiative emission response (to warming?).

Figure 4: "for two different surface temperatures (indicated)" Not indicated

l. 177: The lapse-rate constraint… please be more precise: constant lapse rate?

l. 186:-187. As a non native English reader, I find this sentence hard to understand. Suggestion:… R than it would if R were held fixed/would remain fixed

l. 192 the the

l. 200 is is

l. 212: hte => the

l. 260. Fig.3 => Fig.3 and 6

l. 299 Kluft et al. (2019, 2021) => (Kluft et al.(2019, 2021)

l. 299: not only quantitative, but also quantitative =>  not only qualitative, but…

l. 302: The first is to replace the source function… "=> Planck source function

l. 381: "In the cold regime" ; add "($\eta <1$)"

l. 393: "of the fixed albedo Scs" => of the fixed albedo climate sensitivity Scs

l. 402: "calculate F" => calculate the forcing F

l. 420: "the forcing masking fraction fCO2" add somewhere "by clouds"

l. 459: all=>call

l. 464-465: problems with ,

---

## Author Response (AR2)

**REPLY TO COMMENTS AND REVIEWS OF "A COLORFUL LOOK AT CLIMTE SENSITIVITY"**

BJORN STEVENS AND LUKAS KLUFT

**General Remarks**

The paper has been through a second round of review, with one re-review (rev 1) and one new review (rev2). The main issues raised remain related to style more than to substance – specifically how much to review, how to best present complications introduced by $CO_2$, and ambiguity related to terminology (spectral masking). In the revised manuscript we undertook major changes. Specifically, and as detailed further in the itemized reply to the reviewers, we have made the following changes:

(1) We dropped our attempt to introduce our model axiomatically, for a more direct approach, which helps us be more precise in our introduction of terminology (spectral masking, e.g., Rev2), also through the introduction of schematics.

(2) To better emphasize that our goal is not an elegant asymptotic limit of radiative transfer, but rather a simple model that helps understand the physics of complex radiative transfer calculations, we now discuss our model as a heuristic.

(3) We incorporated the effects of $CO_2$ more naturally (Rev 1), rather than introducing it after the fact as a correction. Incorporating these effects result in less of a moderating effect than the earlier literature, something that is due (mostly) to simplifications in our model, but is also sensitive to the assumption of an unrealistically cold (and thin) stratosphere in models with a more complete treatment of radiative transfer.

(4) Including $CO_2$ encouraged us to develop our ideas less directly around Simpson's law, and more by emphasizing what controls the emission at any given wave number, and how the emission temperature of this constituent couples to the surface – Simpson's law is just one reason why emission temperatures might (not) change. This reduced the emphasis on what the reviewers thought was unnecessary repetition.

(5) Throughout we tried to pair back unnecessary material and sharpen the presentation, so that despite the introduction of more explanatory material the manuscript as a whole is not substantially longer.

RC1

**RC1 Major comments.**

(1) Spectral masking: The rationale for recapitulating the work of Koll, Jeevanjee, and others in sections 3.2-4.2 seems to be that there is a unifying concept of "spectral masking" which is then applied to clouds in section 5. The authors define spectral masking in line 167 as emissions which are "invariant of $T_{\mathrm{sfc}}$", and hence don't contribute to $\Lambda$. That is all well and good, and essentially synonymous with earlier literature which referred to such emission as "Simpsonian" (Ingram 2010) or "Ts-invariant" (Jeevanjee and Romp 2018). But then, in section 4.2, the authors apply this concept to $CO_2$ forcing, arguing that where $CO_2$ is optically thick it "masks" the surface. But emission from such CO2-dominated wavenumbers is not invariant to warming the way H2O-dominated wavenumbers are, as the authors acknowledge, so it seems that the authors' definition of spectral masking does not apply here. The authors seems to be using the term rather loosely, and in some ways synonymously with simply "large optical depth" or "low surface transmissivity".

   We attempted to address this ambiguity through the more direct approach, using Eq. (3), the schematic, and by distinguishing between what we call invariant and sensitive emitters. The main point is to see the control of emissions as being the weighted response of two dominant emitters with the transmissivity of the one that is more optically thick doing the weighting.

(2) State-dependence of $\Lambda$: A key point of section 4.1 and Figure 5 is that the "Simpsonian response parameter" $\Lambda$ has a significant temperature-dependence. However, the calculation of $\Lambda$ seems to employ several significant approximations, including the neglect of pressure broadening in eq. (4) (more on this below) as well as the neglect of $CO_2$. More realistic calculations of the response parameter show a more muted temperature-dependence (Koll + Cronin 2018, Kluft and Stevens 2021, McKim 2021). The authors acknowledge this in 5.1.2, where they note that the difference in response parameters between $288\,\mathrm{K}$ and $302\,\mathrm{K}$ is $3\times$ for their idealized $\Lambda$, and more like 30% for more realistic $\lambda$ values. There is a big difference between $3\times$ and $1.3\times$. It feels misleading to not revise the narrative of section 4.1, or even the whole paper, in light of this. At the very least the authors should present the results of a more comprehensive calculation (including $CO_2$) in Fig. 5, so that readers can make an informed judgment about the significance of this state-dependence.

   We have reformulated the presentation so that the effect of $CO_2$ is now part of the model and included throughout, rather than as an afterthought (new §3.3 and additional Fig. 6). Hence the moderating effect of $CO_2$ becomes apparent from the start, as recommended by the reviewer. We note (also in the manuscript) that the moderating effect still appears less in our simple model than in more complete calculations ($\Lambda(305\,\mathrm{K}) \approx 0.5\,\mathrm{W\,m^{-2}\,K^{-1}}$ as compared to $\approx 1\,\mathrm{W\,m^{-2}\,K^{-1}}$ in the other mentioned studies. However all of those studies adopted a rather cold stratosphere, which influences the relative weight of the $CO_2$ masking versus wing emission. If we adopt $T_{\mathrm{cp}} = 150\,\mathrm{K}$ as (apparently) used by McKim et al., or Koll and Cronin, we get $\Lambda(305\,\mathrm{K}) \approx 0.65\,\mathrm{W\,m^{-2}\,K^{-1}}$, so only part of the discrepancy comes from our simplified treatment.

(3) Pressure Broadening: Despite the author's contention otherwise, Eq. 4 (and hence the calculations shown in Fig. 5, which as far as I can tell are based on Eq. 4) does not include pressure scaling due to collisional broadening. Certainly the reference mass absorption coefficients shown in Fig. 3 have the effects of collisional broadening baked in. But the question is how these coefficients are scaled when evaluated at pressures different from the reference pressure. A standard approximation is to take the far-wing scaling, which is linear in p, and apply it to all wavenumbers. It is this approximation which was justified in Romps 2022. But Eq. 4 has no $p$-scaling at all, and in this sense neglects pressure broadening.

ARTs calculates the absorption spectra (not the absorption coefficients) as a function of pressure, to include broadening effects from all lines. The spectra would thus vary with $P$ were we to use it to perform radiative transfer level by level. We neglect this and assume an effective absorption spectrum to estimate the optical thickness of the atmosphere. This simple minded approach is motivated by a desire to link the radiative response and forcing to bulk measures of the absorption, rather than trying to compute their effect on radiant energy transfer level by level. Essentially we are saying that the optical depth at the surface, which normally would require integrating pressure scaled absorption coefficients over the atmosphere can be approximated in the sense of the mean value theorem. To assess if we were being too simple minded we repeated our calculations with line shapes taken at $P = 700\,\mathrm{hPa}$ and $T = 270\,\mathrm{K}$ instead of at $P = 850\,\mathrm{hPa}$ and $T = 280\,\mathrm{K}$ leads (as expected) to a more transparent atmosphere, but the effect is small with $\Lambda$ increasing by 2 %.

(4) Derivation of $CO_2$ forcing: Section 4.2 on $CO_2$ forcing is somewhat clearer, but still confusing. Why is it permissible to ignore tropospheric temperatures entirely? How does assuming an isothermal stratosphere imply that there should be no stratospheric adjustment? Couldn't the stratosphere cool uniformly? What is the "absorption feature" mentioned in line 249 – simply the edge of the $CO_2$ band? The integral in Eq. 13 is evaluated in line 254 – how is this done? Numerically using $\tau(\nu)$ calculated from the spectroscopy shown in Fig. 3?

We now introduce a schematic to clarify these issues, and refer to the absorption band of $CO_2$. An isothermal stratosphere accounts for stratospheric adjustment (which renders $CO_2$ to be an invariant emitter when it is optically thick within the stratosphere. As the schematic shows, the emission in the troposphere is not neglected, it just doesn't matter for the estimate of the forcing as the area of the parallelogram is given by its 'height' (new Fig. 8) which depends on $T_{\mathrm{sfc}} - T_{\mathrm{cp}}$. This schematic is not so different from what has been used in the past, i.e., in the Jeevanjee et al., study, and to offset some of the repetition it introduces we have tried to pair back our discussion of the forcing, focusing more on how the now familiar way of thinking is manifest in our heuristic.

(5) Derivation of Equation 5: Eq. 5 appears rather abruptly after the discussion of spectral masking, and may confuse readers. In addition to Jeevanjee et al. 2021a, the authors could reference Eq. 5 in the SI (not main text) of Koll and Cronin 2018, which derives the authors' Eq. 5. This derivation is short but insightful, and readers might benefit from seeing it reproduced here.

The (now) Eq. (4) should now follow more directly from the explicit formulation Eq. (3), but the reviewers suggestion to also reference Koll and Cronin has also been adopted.

**RC2**

**RC2 Major comments.** Many of the major comments are related to what we think was a common deficiency in the previous version of the manuscript. This motivated us to be more explicit in the introduction of our model,. This lead not only to the new Eq. (3) but also the introduction of the schematic, Fig. 1.

The minor comments raised in RC2 have also all been addressed, either by adding information that was missing, or reformulating sentences that were unclear. Here we note that there is a tradition of using 'vapor' to denote a condensible 'gas', one whose temperature is below its critical temperature.

(1) I found the word masking inappropriate. A fundamental difference between radiative exchanges in the SW and LW domains is that in the first case a screen (at room temperature) can actually mask radiation, whereas in the LW domain, if a screen absorbs (and therefore masks) radiation, then it also emits some. The vocabulary used must be consistent with this fundamental difference. The use of "cancelling", or probably better "cancelling changes" for example, might be more appropriate because it is not commonly used for visible radiation.

   In addition to the general comment above, wherein the simple model has been introduced to help make clear why we use this language, we also discussed this with the reviewer, and have modified or qualified our usage of the term to avoid any possible confusion.

(2) statement S3 is always true. I believe the authors are referring here to the Simpson effect, or one of its consequences, where the temperature of the atmosphere also varies. This statement requires some additional hypothesis, mainly: the H2O is the only absorber and its optical thickness is large. This does not hold for the CO2 or for clouds. The discussion after Eq. 5 needs also to be revised.

   This has been reformulated to identify it as a form of Beers law, which is recovered when the emission temperature of the masking constituent doesn't change (so the source term that emerges in Schwarzschild's generalization of Beer) vanishes). This is also addressed by adding a reference to the derivation by Koll and Cronin, as suggested by Reviewer 1.

(3) The masking for CO2 and for clouds is very different from the masking by H2O, and I think two different names should be used. For CO2 and clouds, the key point is that they drive the emission temperature, and their effect depends on the difference between the emission temperature when they are not there and when they are present. For H2O, the masking refers to the fact that the emission temperature does not change when the atmospheric temperature changes. Significant parts of the CO2 and the clouds sections need therefore to be modified

   We made several changes to address this point. In particular we try to differentiate between our model, and the spectral masking it implies. We also no longer refer to unmasking. Instead we now talk about restoring the spectral response to warming.'

(4) The value of 275K is present in various places. A variable name, instead of a value, with its interpretation will be welcome.

In an effort to address this comment we have considerably reformulated the presentation of the forcing. Now we introduce an emission temperature, $T_W$ that depends on $\kappa_{\nu,\mathrm{v}}$ and show how to calculate it, when using our heuristic to estimate forcing. We also better separate our heuristic, which is colorful, from the band-averaged approach used int he recent literature, which then uses a band averaged value of $T_W$.

(5) line 225 The value in McKim et al. 2021 has been obtained with an atmosphere with CO2. The comparison is not relevant here but may be relevant later.

This is now addressed through the inclusion of $CO_2$ in our model.

(6) The status of section 4.3.1 is not clear for me as it mainly repeat previous developments. The goal is to present a synthesis?

This has been substantially reformulated. The material that shows how our heuristic allows one to write an expression for the climate sensitivity in terms of basic physical constants, with no real empiricism, and how this proves quite informative. The simplification of this to a form that one could have pieced together from the literature is also pointed out, but is no longer the main point of emphasis..

(7) The 'polar' section (5.1.4) presents new and interesting points, especially the 'polar radiative paradox'. However, the authors should better introduce the goal of this section and make clear that some key assumptions made previously, mainly relevant for the tropics, are not valid for the polar regions.

We reformulated the introduction to this section in light of the reviewer comments. We now write:

*This is less of a paradox when one considers the differences between the poles and the tropics, whether it be by virtue of surface albedo changes, or the decoupling of the polar surface from the polar atmosphere. Here we point out the potential for clouds to also cause a differentiated response of the cold poles, versus the warm tropics, to warming.*

---

## Author Response (AR3)

**REPLY TO COMMENTS AND REVIEWS OF "A COLORFUL LOOK AT CLIMTE SENSITIVITY"**

BJORN STEVENS AND LUKAS KLUFT

**General Remarks**

The reviewer is thanked for his careful reading. As was the case previously the suggestions were thoughtful and constructive and motivated us to considerably revise the presentation of the section on cloud effects, and add more terminological precision throughout.

Specifically, and as mentioned in the itemized reply to the reviewers, we have made the following changes:

(1) We added tables of symbols.
(2) We reorganized §5 to better structure the presentation of the cloud effects and the derivation of the all-sky sensitivity.
(3) We simplified the notation throughout.

We include the difference file (as requested) to more specifically document the nature and extent of the changes.

**RC1**

**RC1 Major comments.**

(1) Spectral masking: The rationale for recapitulating the work of Koll, JeevanjeeWhile the work on all aspects of radiation in the LW is very detailed, the statements concerning changes in the cloud fraction on the one hand, and what is happening in the SW domain on the other, are treated superficially. The small change in cloud fraction (Myers et al., 2021; Vogel et al., 2022) only concerns tropical clouds. While the change in the liquid water content of clouds has little influence in the LW (and is therefore not discussed in this manuscript), it can have a much greater influence in the SW. For example, in mid-high latitudes, the SW effect of clouds is very different depending on how the water-ice transition is treated. In addition, section 5.4 is a bit of a "kitchen sink" in its current state. I think that part of this section should be elsewhere, in one (or more) section where the contribution of clouds to the 3 quantities, forcing, response, sensitivity would be formalised (see also my comment below on section 5.1). I understand the authors' desire to highlight the paradoxical nature of the role of clouds, but in this case their should consider a wider range of possible values and to be more explicit about what they have firmly established, what they 'roughly' estimate.

This comment motivated the substantial restructuring of §5, where more specifically we followed the reviewer's suggestion (also below) to better organize and formalize the various discussions of forcing, response and sensitivity. We also took care to ensure we

emphasized when and where we were assuming clear-sky quantities, often by adding this to section headings to increase its prominence (e.g., §5.21 and to differentiate between things we demonstrate and things we 'crudely approximate' (phrase taken from our revised text.

(2) I feel some text is missing to explain how to read section 5.1 (clouds). Currently this section discusses the role of clouds on (1) forcing, (2) response (3) sensitivity, but this clarification is not done and only the "response" part is well structured. The effect of clouds on forcing is first presented very crudely (336-342), and then better formalised in the "polar" section, although this formalisation remains incomplete. For example, the term fCO2 is much discussed, but it is not precisely defined, and we do not know exactly where and how it comes into play. Why not having a section on "forcing" where all this would be clearly presented? The same for the sensitivity part, quickly and with little justification mentioned lines 343-344, then developed section 5.1.4 but without saying it explicitly.

We have more precisely defined the various cloud quantities, and addressed the suggestion of restructuring as noted above.

(3) In addition I'm still confused by the section on polar amplification. Based on results already presented in the manuscript, this section starts by explaining that the purely radiative sensitivity should be very high at the poles. Finally, using the equations themselves, in clear sky conditions, the authors find that the purely radiative sensitivity at the poles is low after all (Figure 9). So, what?

The last sentence of this section was confusing as it seemed to unravel the main point which was carried in the previous sentence. We have revised the ending to emphasize the abilty of clouds to greatly amplify the sensitivity of the poloar regions.

(4) Specifc Comments in Text Order and some details

All of the reviewer's comments were helpful, clarified the manuscript and were addressed as suggested.